# Cardiopatch platform enables maturation and scale-up of human pluripotent stem cell-derived engineered heart tissues

Ilya Y. Shadrin [1], Brian W. Allen[1], Ying Qian[1], Christopher P. Jackman[1], Aaron L. Carlson[1], Mark E. Juhas[1] & Nenad Bursac[1]

Despite increased use of human induced pluripotent stem cell-derived cardiomyocytes (hiPSC-CMs) for drug development and disease modeling studies, methods to generate large, functional heart tissues for human therapy are lacking. Here we present a "Cardiopatch" platform for 3D culture and maturation of hiPSC-CMs that after 5 weeks of differentiation show robust electromechanical coupling, consistent H-zones, I-bands, and evidence for T-tubules and M-bands. Cardiopatch maturation markers and functional output increase during culture, approaching values of adult myocardium. Cardiopatches can be scaled up to clinically relevant dimensions, while preserving spatially uniform properties with high conduction velocities and contractile stresses. Within window chambers in nude mice, cardiopatches undergo vascularization by host vessels and continue to fire $Ca^{2+}$ transients. When implanted onto rat hearts, cardiopatches robustly engraft, maintain pre-implantation electrical function, and do not increase the incidence of arrhythmias. These studies provide enabling technology for future use of hiPSC-CM tissues in human heart repair.

[1] Department of Biomedical Engineering, Duke University, Durham, NC 27708, USA. Correspondence and requests for materials should be addressed to N.B. (email: nbursac@duke.edu)

Cardiomyocytes (CMs) derived from human embryonic and induced pluripotent stem cells (hPSC-CMs) represent an attractive cell source for drug development and regenerative therapy applications. Given that as many as 1 billion CMs can be lost during human heart attack, significant advancements in cell differentiation, purification, and cryopreservation have been made to enable production of large numbers of highly pure hPSC-CMs[1–4]. These advances offer therapeutic promise, especially when combined with tissue engineering strategies to accelerate hPSC-CM maturation in vitro and to enhance survival, retention, and functional benefits of implanted cells in vivo[5].

Previously, 3D human cardiac tissues have been engineered using scaffold-free cell sheets, synthetic polymer scaffolds, various hydrogels (including collagen, fibrin, and cardiac-derived matrix), and decellularized tissues[6]. Even with the use of electrical and mechanical stimulation, these tissues exhibit function and maturity far inferior to those of adult myocardium, as evidenced by small hPSC-CM size, underdeveloped $Ca^{2+}$ handling[7,8], lack of T-tubules[9,10], absence of H-zones and M-bands[9–12], weak excitability and contractility[7,8,12–18], and slow action potential conduction[10,13,17,19–21].

Importantly, while recent focus has been on cardiac tissue miniaturization for high-throughput drug screening[7,8,12,20], no methods have been developed to generate large, functional heart tissues that would meet the "safety and efficacy" requirements for human cardiac repair. At a minimum, such tissues should: (1) support fast action potential conduction to reduce risk of arrhythmias[22], (2) produce strong contractile forces to aid in mechanical pumping of native heart, (3) be sufficiently large to cover entire infarcted area, and (4) undergo vascularization to promote long-term survival. Thus, there is an immediate need for development of simple, scalable technologies to rapidly engineer highly functional human heart tissues suitable for large animal pre-clinical studies and future clinical applications.

We have recently described use of free-floating dynamic culture conditions to generate miniature, cylindrically shaped heart tissues (cardiobundles) that exhibited near-adult levels of maturation and function[23]. In the current study, we combined our hydrogel-molding methods[24–26] with dynamic culture to develop a versatile in vitro platform for rapid maturation of 3D engineered human heart tissues (cardiopatches) without need for exogenous stimulation. Using multiple hPSC lines, we show that cardiopatches exhibit electrical and mechanical function similar to those of the adult human myocardium. Furthermore, the scalability of the approach is demonstrated by the first-time engineering of cardiopatches with a clinically relevant size (4 × 4 cm), which maintain maturation and functional properties. The cardiopatches also undergo vascularization and maintain electrical function when implanted in dorsal window chambers in nude mice and on the rat epicardium, and show no arrhythmogenesis in vitro or in vivo. Together, our studies suggest the utility of the cardiopatch platform for the future development of next-generation tissue engineering therapies for ischemic heart disease.

## Results

### Cellular makeup and structural maturation of cardiopatches.

Through modification of the WNT signaling pathway[1,2], we differentiated hiPSC monolayers into hiPSC-CMs, with the onset of spontaneous contractions typically between d7 and d9. Following two days of metabolic selection[4] (d10–12) and replating to eliminate non-CMs (Supplementary Movie 1), cells at d15–21 contained $86.3 \pm 0.9\%$ cTnT$^+$ hiPSC-CMs ($n = 66$ independent differentiations; range 71–98% cTnT +; Fig. 1a), with the

remaining ~14% made up primarily of smooth muscle cells and fibroblasts and virtually no endothelial cells (Supplementary Fig. 2). The dissociated cells were encapsulated in hydrogel to form 7 × 7 mm cardiopatches (1 × 10$^6$ cells per patch, Fig. 1b, Supplementary Fig. 1). After 3 weeks of free-floating dynamic culture[23], cardiopatches consisted of densely packed, multi-layered sarcomeric α-actinin (SAA)$^+$ CMs surrounded by a layer of vimentin$^+$ fibroblasts (Fig. 1c) and SM22a$^+$ smooth muscle cells (Supplementary Fig. 3A, upper) and lacked CD31$^+$ endothelial cells (Supplementary Fig. 3A, lower). Cardiomyocytes within cardiopatches exhibited organized cross-striations and abundant Connexin-43$^+$ gap junctions (Fig. 1d) and N-Cadherin$^+$ adherens junctions (Fig. 1e). Use of Nkx2.5 to specifically label cardiomyocytes (Supplementary Fig. 4) and Ki67 as a marker of cell proliferation demonstrated a continuous decrease in Ki67$^+$ CMs from $17.0 \pm 0.7\%$ at 1 week to $8.1 \pm 1.1\%$ at 3 weeks of culture, consistent with a progressive exit from the cell cycle typical of cardiac maturation and development[27]. We further immunostained for ventricular (MLC2v) and atrial (MLC2a) isoforms of myosin light chain, the latter of which is expressed in both atrial and immature ventricular myocytes and lost in ventricular myocytes with maturation[28]. With time of culture, percent of dual positive MLC2a + 2v immature CMs was gradually decreased, with MLC2v single-positive CMs reaching 94% by 3 weeks of culture. Together, these data indicated progressive maturation of hiPSC-CMs within cardiopatches and their predominant ventricular specification.

### Functional maturation of cardiopatches.

Cardiopatches exhibited spontaneous macroscopic contractions (Supplementary Movie 2) with rate that gradually decreased from 90 to 120 bpm at 1 week to 30–60 bpm at 3 weeks of culture (Fig. 2a). Simultaneously, the total and specific active forces of contraction increased from $1.2 \pm 0.3$ mN ($2.9 \pm 0.6$ mN/mm$^2$) at 1 week to $3.9 \pm 0.2$ mN ($13.3 \pm 1.0$ mN/mm$^2$) at 3 weeks of culture (Fig. 2a, b), with cardiopatches exhibiting physiological active and passive force–length relationships (Fig. 2c). Passive tension increased with tissue stretch and time of culture, reaching $1.8 \pm 0.1$ mN at 20% stretch (Fig. 2c). At 3 weeks of culture, passive stiffness of cardiopatches ($26 \pm 5.6$ kPa) approximated diastolic cross-fiber stiffness of adult human ventricles (~20–50 kPa[29,30]). Moreover, cardiopatches demonstrated flat to slightly negative active force–frequency relationships (FFRs) with $95 \pm 0.8\%$ and $83 \pm 2.0\%$ of 1 Hz force maintained at 1.5 and 2 Hz stimulation, respectively, (Supplementary Fig. 5A, B). Interestingly, the FFR slopes were higher in cardiopatches with shorter 1 Hz twitch duration (Supplementary Fig. 5C) and, specifically, in those having shorter 1 Hz twitch relaxation (Supplementary Fig. 5D) but not rise (Supplementary Fig. 5E) times. Furthermore, electrophysiological assessment by optical mapping demonstrated uniform action potential propagation with conduction velocity (CV) that increased over 3 weeks of culture (Fig. 2d), reaching values of $25.1 \pm 1.3$ cm/s (Fig. 2e, Supplementary Movie 3), consistent with the increased expression of Cx43$^+$ gap junctions (Supplementary Fig. 3B). Importantly, similar functional properties (±20%) were obtained in cardiopatches made from three additional hESC lines (HES2, H9, and RUES2), indicating high reproducibility of the methodology (Fig. 2f).

We further optimized cardiopatch maturation and function by varying time of switch from early CM differentiation media (RPMI/B27 + insulin, termed 3D RB+, Supplementary Table 1) to our standard 3D culture media (5% FBS, Supplementary Table 1)[24]. Interestingly, cardiopatches that were switched earlier from 3D RB+ to 5% FBS media, exhibited lower active and specific forces (Fig. 2g, h), but higher CVs (Fig. 2i), which were consistent with

an apparent increase in their Cx43 expression (Supplementary Fig. 6A). Simultaneously, use of 5% FBS media for 1, 2, or 3 weeks promoted localization of N-Cadherin to cell–cell boundaries (Supplementary Fig. 6B), prolonged action potential duration (APD, Fig. 2j), and increased non-myocyte coverage at cardiopatch periphery (Supplementary Fig. 6C). Based on these results, for the remainder of the studies we chose an optimal media condition (1 week 3D RB+, 2 weeks 5% FBS) that produced cardiopatches with both high active forces (4.7 ± 0.3 mN, 15.2 ± 0.9 mN/mm$^2$) and fast CVs (25.8 ± 0.8 cm/s).

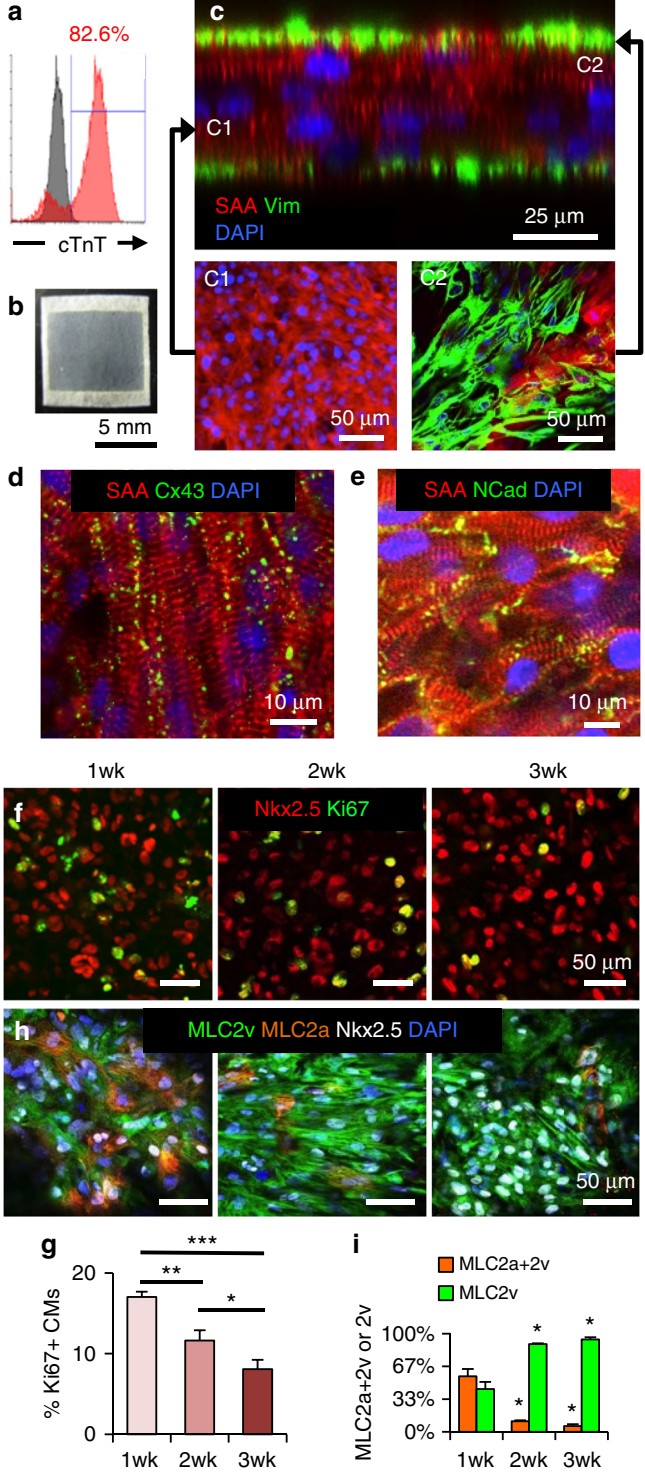

**Enhanced hiPSC-CM properties in low-density cardiopatches.** To provide more room for cell growth and improve the functional output of hiPSC-CMs, we generated cardiopatches with 0.5 million myocytes (0.5MM), half the original cell density. These tissues contained 23% fewer cells per field of view (176.5 ± 10.4 vs. 229.4 ± 16.2 cells; Fig. 3a, b) and were on average 29% thinner (33.4 ± 1.5 vs. 47.4 ± 2.2 μm, Fig. 3c, d) than 1MM patches. The relative cell counts estimated from cardiopatch thickness multiplied by cells per field of view indicated that 0.5MM cardiopatches contained 54% of cells present in 1MM cardiopatches (Fig. 3e), implying preserved input cell ratio after 3 weeks of culture. Furthermore, the relative cell numbers and volumes of 0.5 vs. 1MM tissues (54% of cells in 71% volume) suggested that the cell size in 0.5MM cardiopatches was increased by ~30% in 0.5MM patches. Similar to 1MM cardiopatches, hiPSC-CMs in 0.5MM patches exhibited highly organized sarcomeres and robust electromechanical coupling (Fig. 3f) across entire thickness of the tissue (Supplementary Movie 4), with N-cadherin junctions appearing to localize at cell ends (Fig. 3f, left). Consistent with normal postnatal heart development[31], the Cx43 distribution lagged behind observed N-cadherin polarization, remaining uniform (Fig. 3f, right) and most resembling gap junctional distribution seen in 1–5-year-old human hearts[31]. In addition to cross-striated distribution of cardiac troponin T (cTnT, Supplementary Fig. 7A), strong expression of mature cardiac troponin I (cTnI) (Supplementary Fig. 7B), known to replace immature slow-skeletal TnI (ssTnI) during cardiac development[32], further indicated advanced structural maturation of hiPSC-CMs.

From a functional standpoint, 0.5MM cardiopatches produced 88% of the active force of 1MM patches (5.2 ± 0.2 vs. 5.8 ± 0.2 mN; Fig. 3g), but given their smaller cross-sectional area, yielded significantly higher specific forces (22.4 ± 0.9 vs. 17.7 ± 0.7 mN/mm$^2$; Fig. 3g). Several cardiopatches exhibited active forces >7 mN and stresses >30 mN/mm$^2$ (Supplementary Fig. 8A, Fig. 3g) that were in the range of the 25–44 mN/mm$^2$ values measured for adult human myocardium[33,34]. Furthermore, contractile force production per input CM[24] was nearly two-fold higher in 0.5MM cardiopatches (11.9 ± 0.5 vs. 6.7 ± 0.3 nN per cell; Fig. 3g), while twitch kinetics were significantly faster (rise time of 98.0 ± 2.2 vs. 107.6 ± 2.4 ms; Supplementary Fig. 9A–C), suggesting a more mature contractile apparatus. Consistent with these findings, 0.5MM cardiopatches also showed increased velocity of action potential propagation (28.5 ± 1.0 vs. 25.2 ± 1.1 cm/s, Fig. 3h), with multiple patches having CVs over 40 cm/s (Supplementary Fig. 8B, Fig. 3h), i.e., comparable to an average CV of 46.4 cm/s recorded in adult human ventricles[35].

**Fig. 1** Structural characterization and maturation of hiPSC-CM cardiopatches. **a** Representative flow cytometry histogram from hiPSC-CMs after 20 days of differentiation. **b** Photo of a 7 × 7 mm hiPSC-CM-derived tissue patch (human "cardiopatch") surrounded by a Cerex® frame. **c** Representative cross-sectional confocal image of 3-week-old cardiopatch demonstrating several layers of densely packed sarcomeric α-actinin (SAA)$^+$ hiPSC-CMs (C1) surrounded by a layer of vimentin (Vim)$^+$ fibroblasts (C2). **d**, **e** Representative confocal images of connexin-43 (Cx43, **d**) and N-Cadherin (NCad, **e**) junctions in cardiopatch. **f**, **g** Representative confocal images (**f**) and quantification (**g**) of Ki67$^+$/Nkx2.5$^+$ CMs after 1, 2, and 3 weeks of cardiopatch culture; n = 8/11/12 patches (for 1/2/3 week) from four differentiations; *p = 0.037, **p = 0.0033, ***p < 0.0001, post-hoc Tukey's test. **h**, **i** Relative fractions of myosin light chain 2v (ventricular) and 2a + 2v (atrial/early ventricular) positive hiPSC-CMs within cardiopatches cultured for 1–3 weeks; n = 7/5/4 patches (for 1/2/3 week) from four differentiations; *p = 0.0002 vs. 1 week, post-hoc Tukey's test. Data are presented as mean ± SEM. Scale bars **b** 5 mm; **c** 25 μm (C1–C2, 50 μm); **d**, **e** 10 μm; **f–h** 50 μm

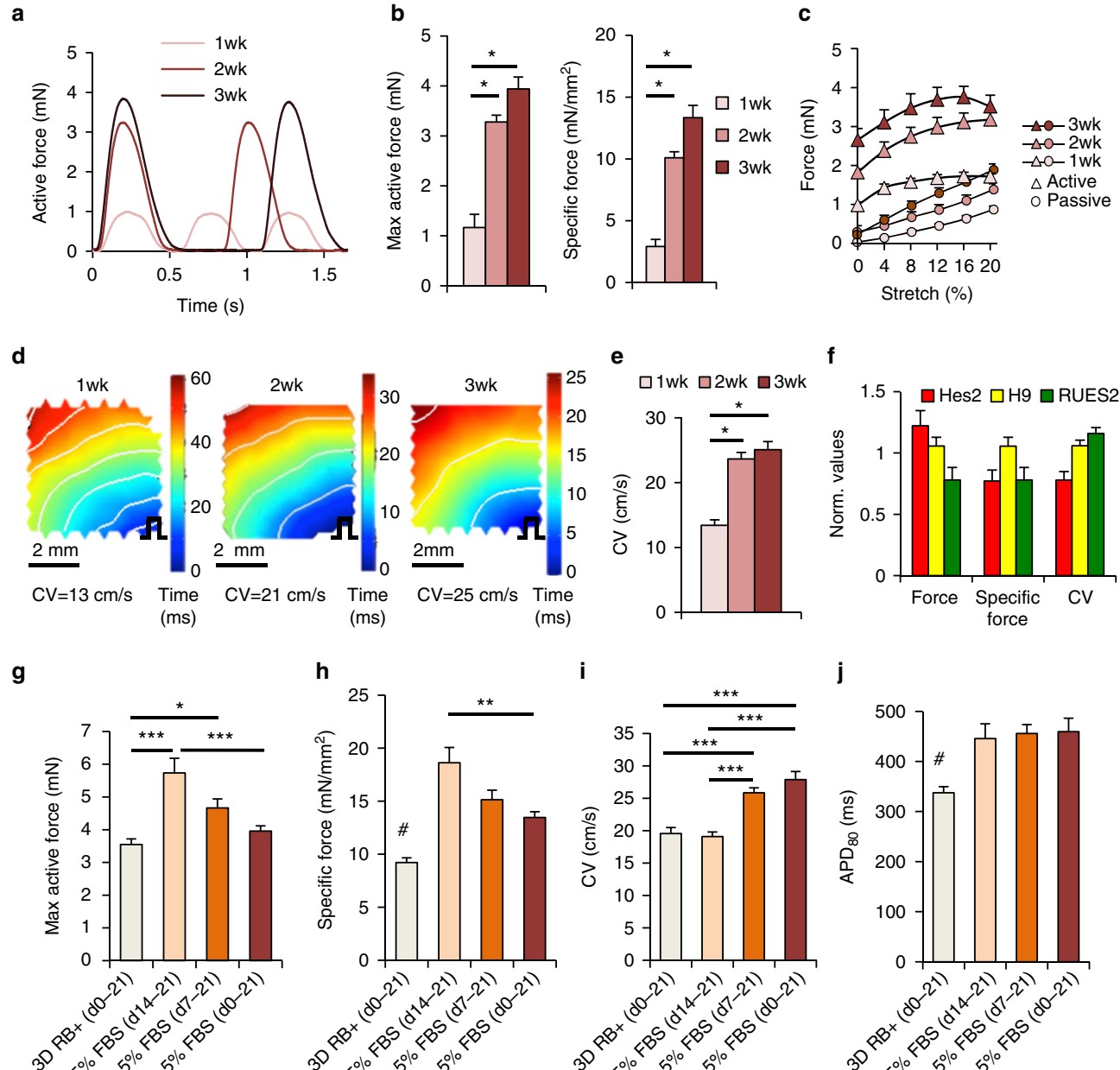

**Fig. 2** Functional assessment and maturation of human cardiopatches. **a** Representative isometric contractile (active) force traces of spontaneously beating hiPSC-CM cardiopatches at 1, 2, and 3 weeks of culture. **b** Maximum active force (left) and specific force (right) of 1, 2, and 3-week-old cardiopatches; $n = 6$ patches from two differentiations; *$p < 0.0001$, post hoc Tukey's test. **c** Active and passive force–length relationships of isometrically tested (1 Hz stimulation) cardiopatches at 1, 2, and 3 weeks of culture; same patches as in **b**. **d**, **e** Representative isochrone activation maps (**d**) and average conduction velocities (CV, **e**) during point stimulation from bottom right corner (pulse sign) of cardiopatches at 1, 2, and 3 weeks of culture; $n = 7/7/9$ patches (for 1/2/3 weeks) from two differentiations; *$p < 0.0001$, post hoc Tukey's test. Scale bars **d**, 2 mm. **f** Average active force, specific force, and CV in cardiopatches obtained from three additional hESC lines (Hes2, H9, and RUES2) normalized to those made of hiPSC-CMs; $n = 18/37/22$ patches (for Hes2/H9/RUES2) from five to seven independent differentiations per line. **g**, **h** Maximum active force (**g**) and specific force (**h**) of cardiopatches cultured for 3 weeks in 3D RB + medium or 1, 2, and 3 weeks in 5% FBS medium; $n = 15/14/13/14$ patches (for 3D RB+/1/2/3 weeks in 5% FBS) from five differentiations; *$p = 0.0026$, **$p = 0.0005$, ***$p < 0.0001$, #$p < 0.001$ vs. all other groups, post hoc Tukey's test. **i**, **j** CV (**i**) and action potential duration at 80% repolarization (APD$_{80}$, **j**) in cardiopatches cultured for 3 weeks in 3D RB + medium or 1, 2, and 3 weeks in 5% FBS medium; $n = 15/15/14/15$ patches (for 3D RB+/1/2/3 weeks in 5% FBS) from five differentiations; ***$p < 0.0001$, #$p < 0.002$ vs. all other groups, post hoc Tukey's test. Data are presented as mean ± SEM

Additionally, 0.5MM cardiopatches exhibited a trend toward lower APD$_{80}$ than 1MM patches ($423.7 \pm 10.8$ vs. $444.9 \pm 10.3$ ms, $p < 0.16$, unpaired $t$-test; Fig. 3h), better approximating values measured in human myocardium (350–430 ms for 1 Hz pacing)[36]. Taken together, lowering hiPSC-CM density in cardiopatches significantly improved their electromechanical properties to approach functional parameters of adult myocardium. Furthermore, cardiopatches cultured under traditional static (instead of our regular dynamic) conditions exhibited 4.9-fold reduced contractile forces (1.03 vs. 5.1 mN; Supplementary Fig. 10A, B), ~3-fold lower CVs (8.7 vs. 27.2 cm/s; Supplementary Fig. 10C, D) less organized sarcomeric structure,

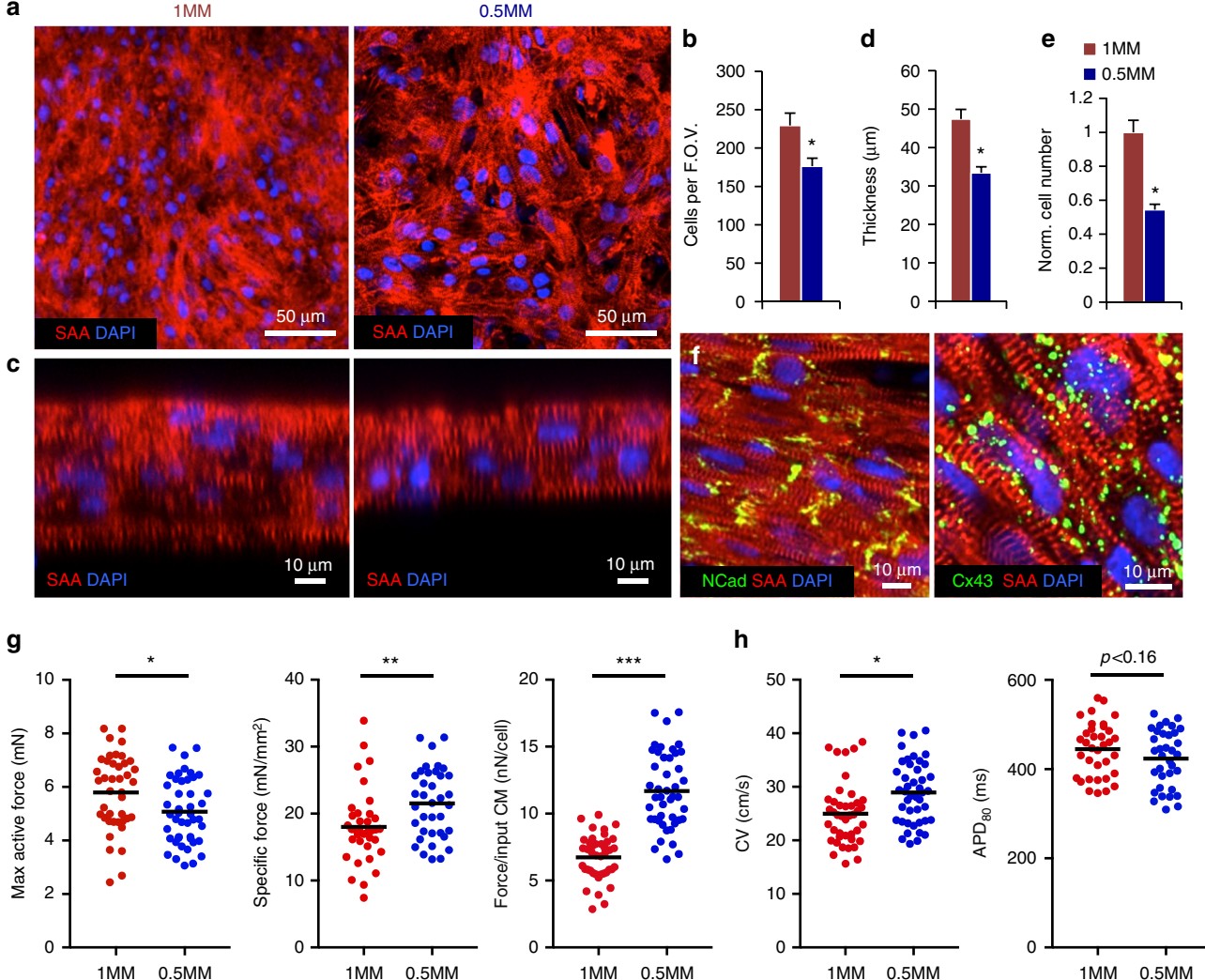

**Fig. 3** Lowering density of hiPSC-CMs improves cardiopatch function. **a** Representative confocal images of 3-week-old cardiopatches generated from 1 million (1MM) and 0.5 million (0.5MM) hiPSC-CMs stained for SAA. Scale bar 50 µm. **b** Quantification of average number of cells per field of view in 0.5 and 1MM cardiopatches using a ×63 objective (2 µm slice); $n = 13/16$ patches (1/0.5MM) from seven differentiations, three to four random fields of view per patch; $*p = 0.0083$, unpaired t-test. **c** Representative optical cross-sections of 1MM and 0.5MM cardiopatches. Scale bar 10 µm. **d** Quantification of cardiopatch thickness made from 0.5 and 1MM cells; $n = 22/25$ patches (1/0.5MM) from 7/11 differentiations, average 2 thickness measurements per patch; $*p < 0.0001$, unpaired t-test. **e** Calculated cell numbers in cardiopatches (normalized to 1MM patch) based on average patch thickness and cells per field of view; $n = 13/16$ patches (1/0.5MM) from seven differentiations; $*p < 0.001$, unpaired t-test. **f** Representative confocal images of 3 week 0.5MM cardiopatches stained for Cx43, SAA and NCad. Scale bar 10 µm. **g** Maximum active force, specific force, and force per input hiPSC-CM in 1MM and 0.5MM cardiopatches; $n = 36/35$ patches (1/0.5MM) from 13 differentiations; $*p = 0.047$, $**p = 0.0021$, $***p < 0.0001$, unpaired t-test. **h** Conduction velocity (CV) and action potential duration at 80% repolarization ($APD_{80}$) of 1MM and 0.5MM cardiopatches; $n = 42/44$ patches (1/0.5MM) from 10 differentiations; $**p = 0.0028$, unpaired t-test. Data in **b**, **d**, and **e** presented as mean ± SEM, while dot plots in **g** and **h** show all data points along with mean value (black line)

and decreased Cx43 expression (Supplementary Fig. 10E), resembling the appearance of 1-week dynamically cultured tissues. Thus, in agreement with our recent study[23], dynamic culture was essential for promoting the advanced functional phenotype of cardiopatches.

**Molecular and ultrastructural maturation of hiPSC-CMs.** We further sought to reveal the molecular and ultrastructural signatures underlying advanced functional maturation of 0.5MM cardiopatches. Based on previous studies comparing gene expression patterns among hPSC-CMs, fetal and adult human myocardium, we identified a panel of 10 cardiac "maturation genes" (Supplementary Tables 3 and 4) with the highest adult:fetal

(up to 14-fold) and adult:hPSC-CM (up to 797-fold) expression ratios. These 10 genes were classified into three groups—structural (*TNNI3*, *MYL2*, *MYOM2*, *MYOM3*), excitation–contraction (E–C) coupling (*CASQ2*, *S100A1*, *PLN*), and metabolic (*COX6A2*, *CKMT2*, *CKM*), thus reflecting key maturation processes in developing CMs. Nine out of the ten genes were found to progressively increase with time of culture, while phospholamban (*PLN*, previously shown lacking expression in hiPSC-CMs[37]) remained steady at high level (55–70% of adult heart levels) (Fig. 4a) between culture weeks 1 and 3. Importantly, gene expression at 3 weeks of culture was 3- to 163-fold higher (median 23-fold) compared to d0, including a 108-fold and 163-fold increase in *MYL2* and *CASQ2* expression (Fig. 4a, Supplementary Fig. 11A). Furthermore, an average 3-fold

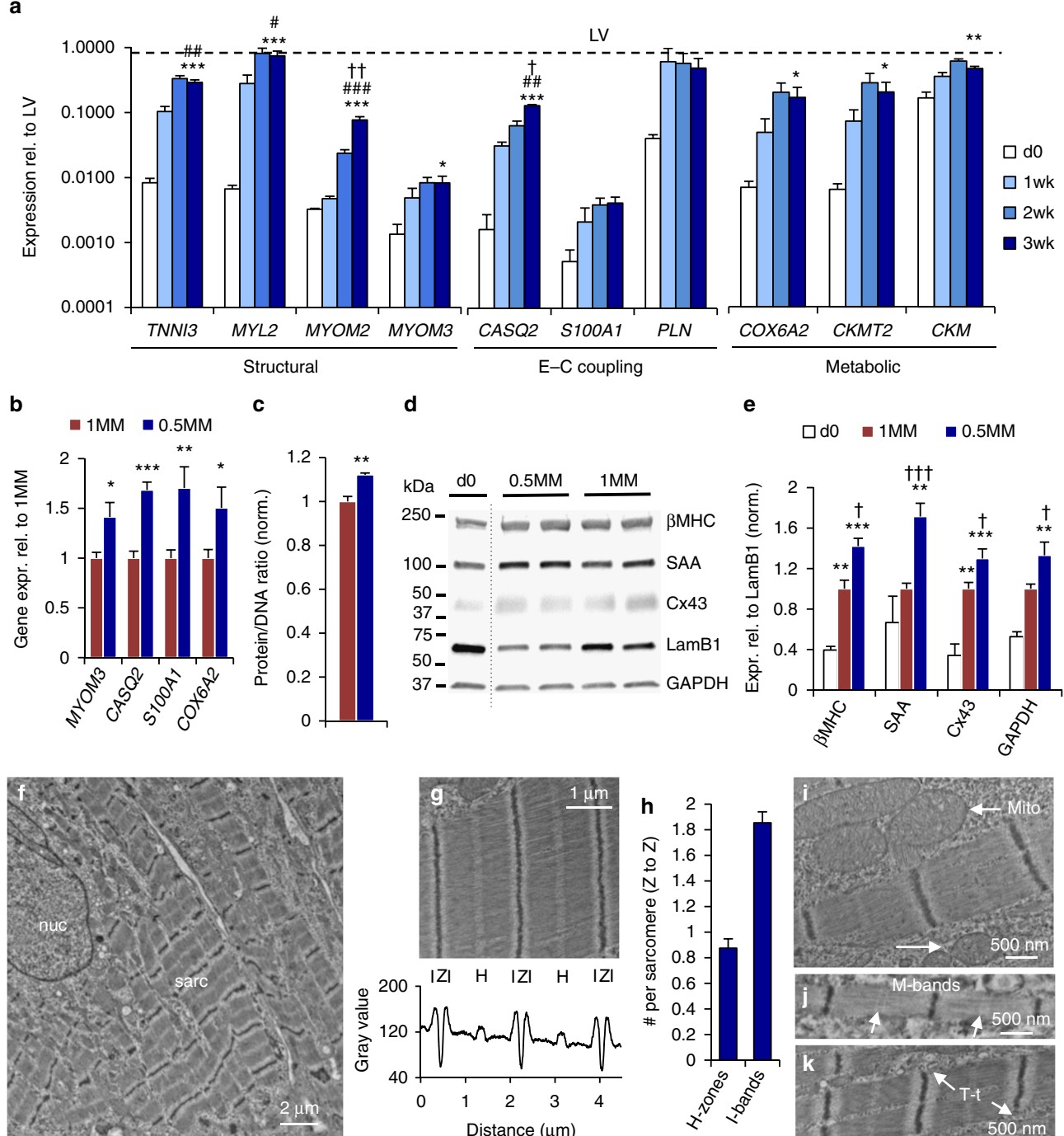

**Fig. 4** Increased cell maturation and size in low-density cardiopatches. **a** Relative expression levels of 10 maturation genes in d0, 1 week, 2 week, and 3 week cardiopatches compared with those in adult human left ventricles (LV). Linear regression from d0 to 3 weeks: *TNNI3, MYL2, MYOM2, CASQ2*, $p < 0.0001$; *MYOM3, CKMT2*, $p < 0.014$; *S100A1, COX6A2, CKM*, $p < 0.006$; *PLN*, $p = 0.1081$. $*p < 0.05$, $**p < 0.001$, $***p < 0.0001$ vs. d0; $\#p < 0.023$, $\#\#p < 0.0006$, $\#\#\#p < 0.0001$ vs. 1 week; $\dagger p < 0.037$, $\dagger\dagger p < 0.0037$ vs. 2 week, via post hoc Tukey's tests. For clarity, only 3-week statistical comparisons are shown. **b** Relative gene expression in 0.5MM vs. 1MM cardiopatches; $*p < 0.041$, $**p < 0.0073$, $***p < 0.0001$, post hoc Tukey's tests. **c** Protein/DNA ratio in 0.5MM vs. 1MM cardiopatches; $n = 3/4$ patches (1/0.5MM). **d** Representative western blots for myosin heavy chain-β (βMHC), SAA, lamin B1 (LamB1), GAPDH, and Cx43 in d0 cells and 3-week 0.5MM and 1MM cardiopatches; dotted line indicates lanes spliced from the same gel. **e** Quantified protein levels in d0 cells, 1MM and 0.5MM cardiopatches normalized to nuclear envelope protein LamB1, shown relative to 1MM cardiopatches; $**p < 0.01$, $***p < 0.001$ vs. d0; $\dagger p < 0.05$, $\dagger\dagger p < 0.01$, $\dagger\dagger\dagger p < 0.001$ vs. 1MM, post hoc Tukey's tests. For gene expression studies (**a**, **b**), $n = 6$ patches from two differentiations; for protein studies (**c–e**), $n = 8/10$ patches (1/0.5MM) from three to four differentiations. **f** Representative low-magnification view of cell nucleus (nuc) and surrounding sacromeric structures (sarc) in 3-week-old 0.5MM cardiopatches. Scale bar 2 μm. **g** Localization of I-bands, Z-discs, and H-zones within hiPSC-CM sarcomeres. Scale bar 1 μm. **h** Average number of H-zones and I-bands per sarcomere; $n = 4$ patches from two differentiations, data compiled from a total of 74 random fields of view. **i** Mitochondria (mito) were found positioned alongside CM myofibrils. Scale bar 500 nm. **j, k** Evidence of M-bands (**j**) and T-tubular-like structures (T-t, **k**) in 3-week-old 0.5MM cardiopatches. Scale bars 500 nm. All data are presented as mean ± SEM

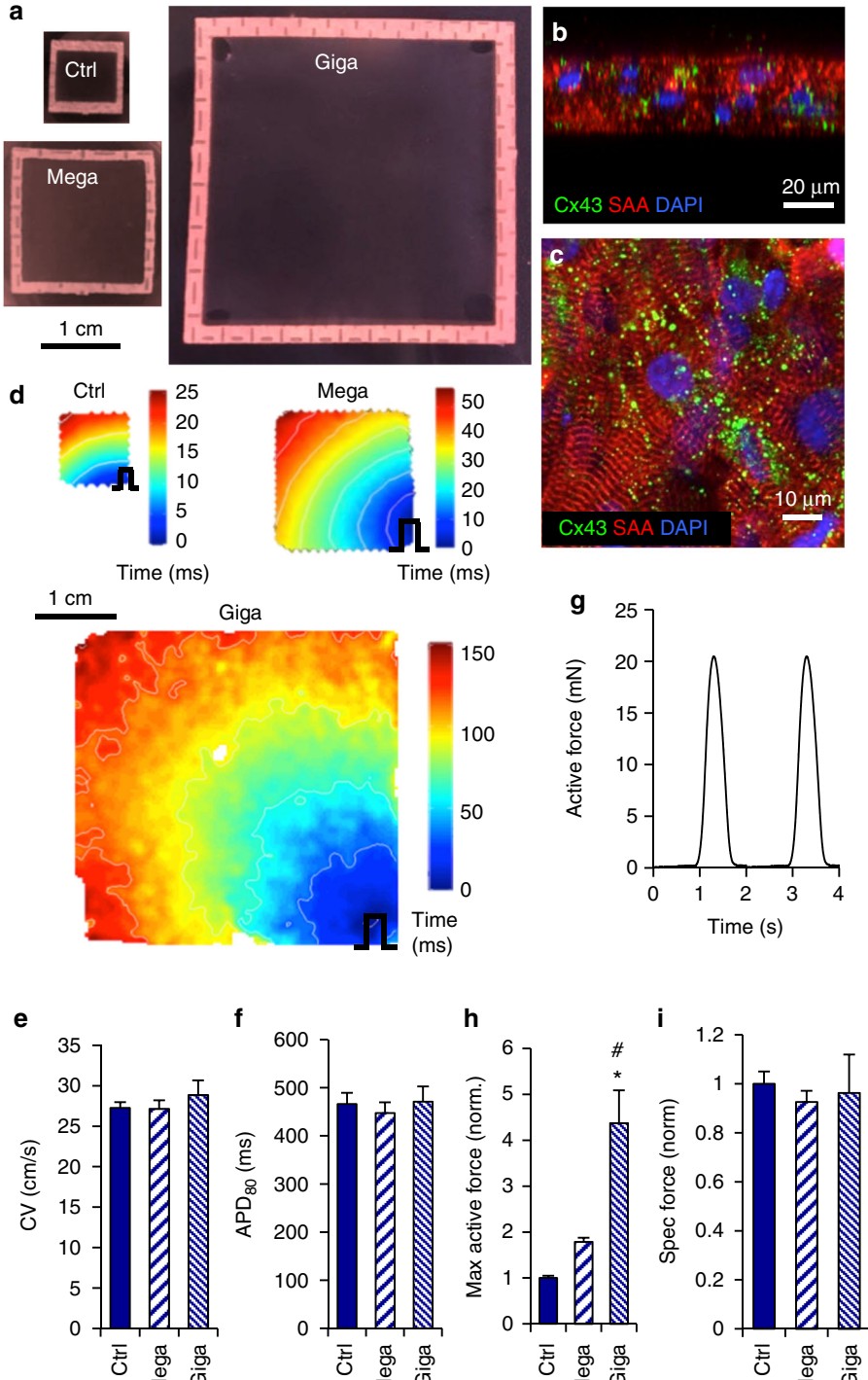

**Fig. 5** Scale-up of cardiopatches without loss of function. **a** Representative images of control (ctrl, 7 × 7 mm), Mega (15 × 15 mm) and Giga (36 × 36 mm) cardiopatches at 3 weeks of culture. Scale bar 1 cm. **b**, **c** Representative confocal images of 3-week-old Giga cardiopatches stained for Cx43 and SAA, as seen in confocal cross-sections (**b**) or in the XY plane in the middle of the patch (**c**). Scale bars 20 μm (**b**), 10 μm (**c**). **d** Representative activation maps of ctrl, Mega, and Giga cardiopatches following point stimulation from bottom right corner (pulse sign). Giga patches were imaged by an EMCCD camera. Scale bar 1 cm. **e** Conduction velocity (CV) in 3-week-old ctrl, Mega and Giga cardiopatches; $n = 11/6/7$ patches (ctrl/Mega/Giga) from three to four differentiations; $p = 0.56$, one-way ANOVA. **f** Action potential duration at 80% repolarization ($APD_{80}$) in 3-week-old ctrl, Mega and Giga cardiopatches; $n = 11/6/5$ patches (ctrl/Mega/Giga) from three to four differentiations; $p = 0.52$, one-way ANOVA. **g** Representative isometric force trace from spontaneously contracting 3-week-old Giga cardiopatch at 16% stretch. **h**, **i** Maximum active forces (**h**) and specific forces (**i**) in 3-week-old ctrl, Mega, and Giga cardiopatches shown relative to Ctrl cardiopatch; $n = 10/6/10$ patches (ctrl/Mega/Giga) from three to four differentiations; *$p < 0.0001$ vs. ctrl, #$p < 0.01$ vs. Mega, post hoc Tukey's tests; (**i**) $p = 0.76$, one-way ANOVA. Data are presented as mean ± SEM

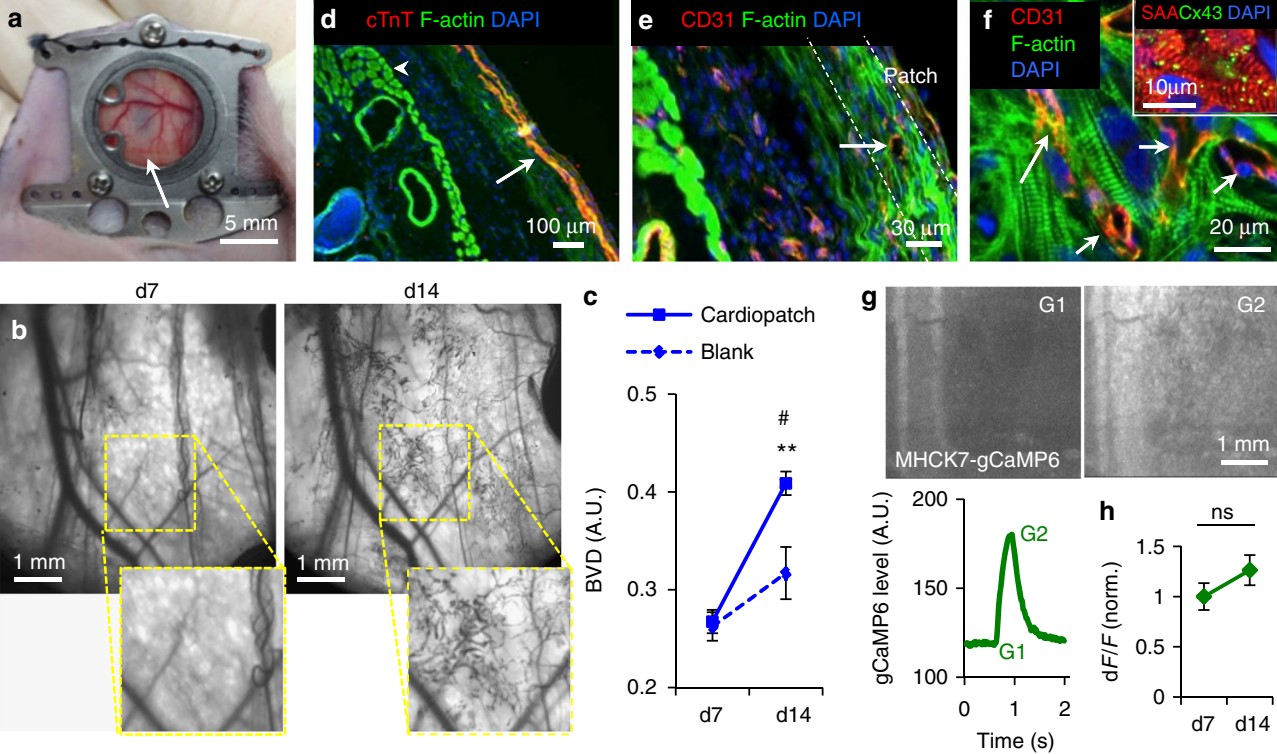

**Fig. 6** In vivo vascularization of cardiopatches. **a** Representative photograph of an implanted 3-week-old cardiopatch (arrow) in a dorsal skinfold window chamber of a nude mouse. **b**, **c** Intravital raw vascularization images (**b**) and quantification of blood vessel density (BVD, **c**) of cardiopatches on d7 and d14 post implantation relative to blank (Cerex® frame-only) controls; $n = 18$ mice (15 cardiopatches from 3 differentiations, 3 blank controls); repeated measures ANOVA: time F-ratio 34.8 ($p < 0.0001$), patch×time interaction effect F-ratio 6.81 ($p < 0.02$); for cardiopatch at d14: **$p < 0.0001$ vs. cardiopatch at d7, #$p < 0.027$ vs. blank at d14, post hoc Tukey's tests. **d**, **e** Representative cross-sections of explanted cardiopatches after 2 weeks in vivo stained for F-actin, cTnT (**d**, arrow showing the cardiopatch) and CD31 (**e**, arrow showing a capillary lumen). **f** Representative en face image of cardiopatch explanted 2 weeks post implantation stained for F-actin and CD31 (arrows pointing to vessel lumens), and SAA and Cx43 (inset). **g** Representative fluorescence images and time trace of gCaMP6 signal during cardiopatch spontaneous activity 1 week after implantation; **f**, cardiopatch frame. **h** Ca²⁺ transient amplitude assessed as relative gCaMP6 fluorescence (dF/F) in cardiopatches at d7 and d14 following implantation; $n = 12$ patches from two differentiations; $p = 0.2$, paired t-test. Data are presented as mean ± SEM. Scale bars **a** 5 mm; **b** 1 mm; **d** 100 μm; **e** 30 μm; **f** 20 μm (inset 10 μm); **g** 1 mm

increase in the expression of 9 out of 10 maturation genes (with 10-fold higher *MYOM2* expression) was found compared to age-matched monolayers (Supplementary Fig. 11B), while significantly higher expression of *MYOM3*, *CASQ2*, *S100A1*, and *COX6A2* (encompassing genes from each all three maturation categories) was found compared to 1MM patches (Fig. 4b). Consistent with the morphometric analysis, additional indices of cell size, such as the total protein per DNA (Fig. 4c) and the total GAPDH protein per nuclear envelope protein LamB1 (Fig. 4d, e, Supplementary Fig. 19A), were 13 and 33% higher in 0.5MM than 1MM cardiopatches, respectively, providing further evidence for increased CM size in less dense tissues. Moreover, higher expression of phosphorylated Akt, with minimal changes in mTOR and AMPK phosphorylation (Supplementary Figs. 12 and 19B), suggested Akt-mediated signaling as a mechanism for the increased CM size in 0.5MM patches. Along with the observed gene expression changes and CM hypertrophy, 1.7, 1.4, and 1.3-fold higher expression of SAA, βMHC, and Cx43 proteins, respectively (Fig. 4d, e, Supplementary Fig. 19A), further supported the finding of enhanced CM maturation in 0.5MM vs. 1MM cardiopatches.

At the ultrastructural level, hiPSC-CMs in 3 week 0.5MM cardiopatches exhibited highly regular and organized sarcomeres (Fig. 4f, Supplementary Fig. 13A) with prominent central H-zones and two distinct I-bands adjacent to Z-discs (0.88 ± 0.07 H-zones, 1.85 ± 0.08 I-bands per sarcomere; Fig. 4g, h), more frequently than in other engineered human cardiac

tissues[9,11–13,38] and approaching the reproducible Z-I-H-I-Z pattern seen in adult human cardiomyocytes[39]. Consistent with functional results, adjacent cardiomyocytes were interconnected via desmosomes and gap junctions (Supplementary Fig. 13B), while internal myofibrils were surrounded by abundant mitochondria with well-developed cristae (Fig. 4i). Notably, some cells (<5%) demonstrated clear M-bands in the center of the H-zones (Fig. 4j), a hallmark of mature sarcomeres that in hiPSC-CM monolayers was previously seen only after 360 days of culture[40]. A fraction of the hiPSC-CMs (<5%) also exhibited T-tubule-like structures adjacent to Z-discs (Fig. 4k), with additional evidence for T-tubulogenesis coming from upregulated expression of T-tubule-associated proteins[41,42] Caveolin-3 (Cav3) and Junctophilin-2 (JPH2, Supplementary Figs. 14A, B and 19C) and Cav3 accumulation observed in sarcomeres (Supplementary Fig. 14C). While live membrane staining with Di-8-ANEPPS (Supplementary Fig. 14D) did not reveal cross-striated T-tubular pattern characteristic of adult cardiomyocytes, these results supported significant maturation of the E–C coupling machinery during 3-week cardiopatch culture. Taken together, these data showed that lowering cell density in cardiopatches promoted hiPSC-CM hypertrophy, molecular and ultrastructural maturation to approximate several features of adult myocardium.

**Scale-up of cardiopatch size.** Having established a robust 3D culture system for maturation of hiPSC-CMs, we asked whether

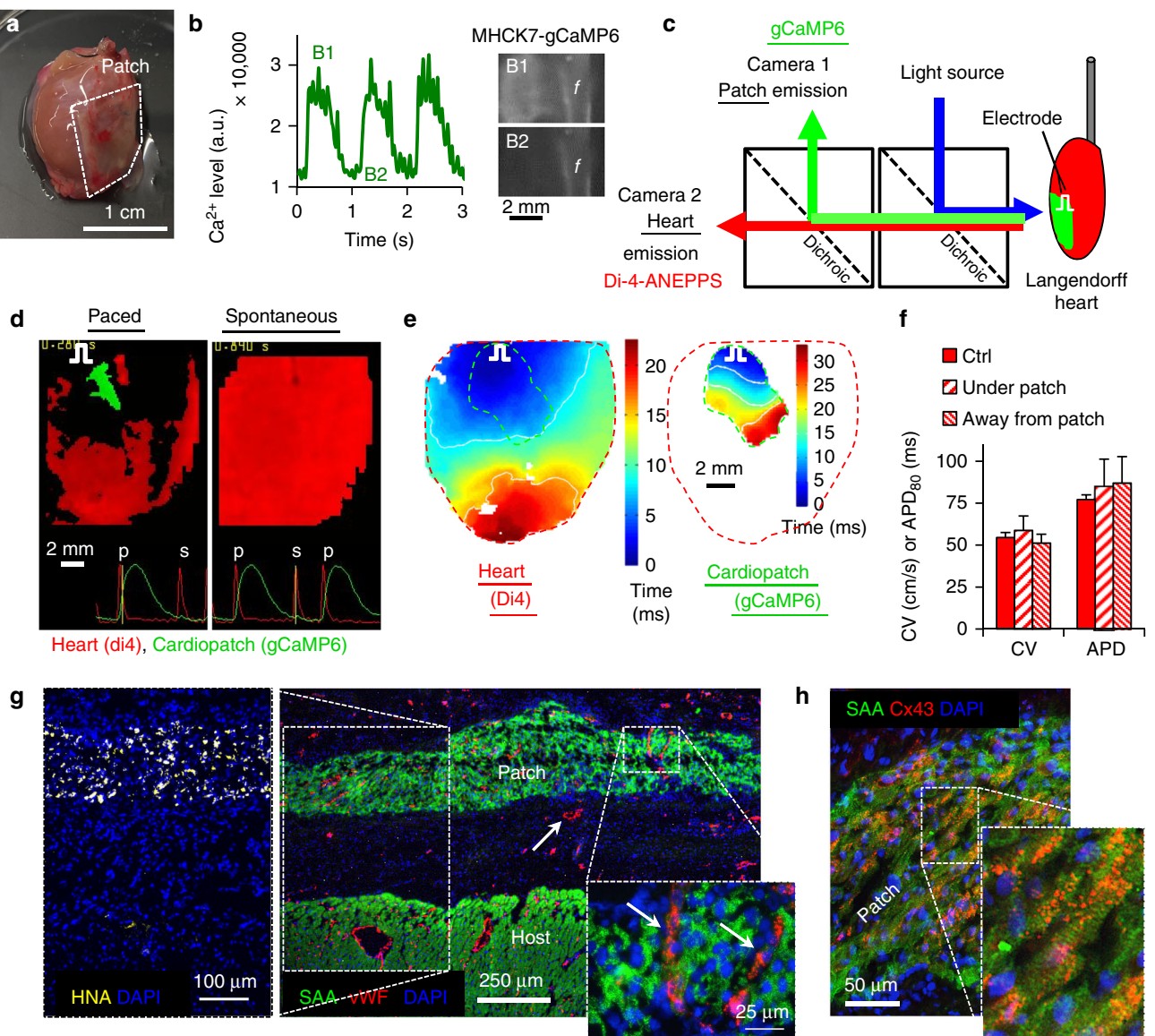

**Fig. 7** Epicardial implantation and ex vivo assessment of cardiopatches. **a** Representative image of cardiopatch 3 weeks following implantation onto nude rat epicardium; f, cardiopatch frame. **b** MHCK7-gCaMP6 flashes in implanted cardiopatches following direct stimulation by a platinum electrode; f, cardiopatch frame. **c** Schematic of a setup for dual optical mapping of gCaMP6-reported $Ca^{2+}$ transients in implanted cardiopatches and Di-4-ANEPPS-reported transmembrane voltage in Langendorff-perfused rat hearts. **d** Representative snapshots from movies of $Ca^{2+}$ transients in cardiopatches (green) and membrane voltage in the heart (red). Traces at the bottom show representative gCaMP6 (green) and Di-4 (red) signals from a single recording channel with yellow line denoting point in time corresponding to the instant of the movie snapshot. Pulse sign denotes location of stimulus electrode; p, paced; s, spontaneous. **e** Representative isochronal maps of action potential propagation during direct point electrode stimulation (pulse sign) of implanted cardiopatch (black dashed outline) and underlying rat myocardium (red dashed outline). **f** CV and APD of host epicardium optically recorded in control conditions (no patch), under implanted cardiopatch (under patch), and remote from implanted cardiopatch (away from patch); $n = 6/3/3$ (control/under/away). **g**, **h** Representative cross-sections of cardiopatch 3 weeks after implantation onto rat epicardium stained for SAA, von Willebrand Factor (vWF), and human nuclear antigen (HNA, **h**) and SAA and Cx43 (**h**). Data are presented as mean ± SEM. Scale bars **a** 1 cm; B1–2 2 mm; **d**, **e** 2 mm; **g** main: 250 µm, left inset: 100 µm, right inset: 25 µm; **h** 50 µm

the 0.5MM 7×7 mm cardiopatches could be scaled up to a clinically relevant area without loss of advanced electrical and mechanical function. We thus modified our hydrogel engineering approach to generate 15×15 mm (Mega) and 36×36 mm (Giga) cardiopatches using 2 and 10 million CMs, respectively (Fig. 5a, Supplementary Fig. 15). After 3 weeks of free-floating dynamic culture (Supplementary Movie 5), scaled up cardiopatches exhibited spontaneous, synchronous contractions (~30–60 bpm) that could be overridden by higher rate electrical pacing (Supplementary Movie 6). Similar to control 7×7 mm constructs

(Figs. 1D and 3F), Giga cardiopatches were 34.8 ± 1.8 µm thick (Fig. 5b) and densely populated by cross-striated and robustly coupled hiPSC-CMs (Fig. 5c). Despite significant scale-up in size, both Mega and Giga cardiopatches exhibited similar CVs (Mega 27.2 ± 1.1 cm/s, Giga 28.9 ± 1.8 cm/s) and APDs (Mega 447 ± 22 ms, Giga 471 ± 31 ms) to those of control constructs (Fig. 5e, f), indicating preservation of electrical phenotype. In addition, scaled-up cardiopatches supported spatially uniform action potential propagation throughout the entire area (Fig. 5d, Supplementary Movie 7), and did not initiate re-entrant

arrhythmias during burst pacing. Twitch forces in cardiopatches increased with increase in tissue size, averaging for this set of experiments to $4.6 \pm 0.6$, $9.4 \pm 1.0$ to $17.5 \pm 1.1$ mN for control, Mega and Giga patches, respectively (reaching >20 mN in Giga cardiopatches, Fig. 5g), with specific forces that were similar for all groups ($19.2 \pm 2.2$, $19.4 \pm 2.1$, and $17.0 \pm 0.8$ mN/mm$^2$, respectively). Similarly, for cardiopatches derived from the same input cells, relative increase in twitch force with tissue scale-up (Fig. 5h) yielded comparable specific forces (Fig. 5i), indicating successful preservation of cardiopatch contractile strength. Together, these results demonstrated the first-time engineering of large, highly functional, non-arrhythmogenic human heart tissues.

**Cardiopatch vascularization and function in window chambers.** We then assessed the ability of cardiopatches to vascularize and remain functional in vivo using our previously described dorsal window chamber assay in nude mice[43,44]. The 3-week-old 0.5MM cardiopatches ($7 \times 7$ mm) were implanted into dorsal window chambers and imaged through a glass coverslip overlying the implant (Fig. 6a). Analysis of intravital images (Fig. 6b, Supplementary Fig. 17) demonstrated progressive implant vascularization with a 1.6-fold increase in blood vessel density (BVD, Fig. 6c) between 7 and 14 days post implantation, significantly more compared to cell-free, frame-only controls (Fig. 6c). By 2 weeks post implantation, blood flow through the newly formed vessels was clearly visible (Supplementary Movie 8), while cross-sectional stainings revealed intact cTnT$^+$ cardiopatches (Fig. 6d) containing CD31$^+$ capillary lumens and Cx43-coupled striated cardiomyocytes (Fig. 6e, f). Considering that implanted patches contained no endothelial cells (Supplementary Fig. 3A, lower), observed vascularization in vivo likely originated from the host vessel ingrowth. To assess cardiopatch functionality in vivo, hiPSC-CMs were transduced with an MHCK7-gCaMP6 lentivirus[45] and spontaneous Ca$^{2+}$ transients were intravitally monitored through the chamber window. Implanted cardiopatches demonstrated strong, synchronous Ca$^{2+}$ flashes (Fig. 6g, Supplementary Movie 9) with amplitudes (d$F$/$F$) that remained stable during 2 week period (Fig. 6h). Overall, these results indicated that in vitro engineered avascular cardiopatches can undergo progressive vascularization and maintain functionality in vivo.

**Functional analysis of epicardially implanted cardiopatches.** To further validate the findings from the dorsal window chambers in a more relevant implantation environment and support translational prospects of our approach, 2-week-old 0.5 MM cardiopatches ($7 \times 7$ mm) were implanted onto the nude rat epicardium and assessed 3 weeks post implantation (Fig. 7a). Upon extraction and Langendorff perfusion of patch-implanted hearts, 10 of 11 cardiopatches exhibited spontaneous and exogenously stimulated (1–2.5 Hz) gCaMP6-reported Ca$^{2+}$ transients (Fig. 7b), indicating successful engraftment and survival. Dual camera mapping (Fig. 7c) of electrical activity in the heart (Di-4-ANEPPS) and cardiopatch (gCaMP6) showed no evidence of graft-host functional coupling since spontaneous or pacing-induced (remote from the cardiopatch) activation in epicardium did not yield Ca$^{2+}$ transients in the patch (Fig. 7d, Supplementary Movie 10). On the other hand, simultaneous stimulation of cardiopatch and underlying epicardium by a point electrode placed near the cardiopatch frame induced electrical propagation in both the patch and the heart (Fig. 7d, e; Supplementary Movie 11). From these studies, CVs in implanted cardiopatches ($18.1 \pm 1.7$ cm/s, Fig. 7e, right) were found to be comparable to pre-implantation CVs ($19.3 \pm 1.0$ cm/s), indicating preserved patch structure and electrical function in the epicardial environment. Furthermore,

cardiopatches did not disturb the propagation pattern of underlying epicardium (Fig. 7e, left), its CV or APD (Fig. 7f), suggesting no adverse paracrine effects from the grafted cells. Consistent with functional results, immunohistological assessment (Fig. 7g) confirmed robust epicardial engraftment of cardiopatches, which contained densely packed cardiomyocytes labeled by SAA and human nuclear antigen (HNA, Fig. 7g, left) and interconnected via Cx43 gap junctions (Fig. 7h). As expected from the lack of electrical coupling, cardiopatches were insulated from the host epicardium by a ~200–300 μm layer of non-cardiac tissue. Similar to window chamber results, host blood vessels were found ingrown in the implanted cardiopatches and between the patch and host myocardium (Fig. 7g, arrows, right).

While the hearts with cardiopatches retained normal CVs and APDs, suggesting no adverse effects on host electrical function, we employed programmed electrical stimulation, ECG recordings, and optical mapping to systematically assess vulnerability to arrhythmias of the patch-implanted vs. control (non-operated, age-matched) hearts ($n = 6$ for each). In 75% of attempts, burst pacing at 2–20 Hz (Supplementary Fig. 18A) failed to produce arrhythmias in both patch-implanted (40/54 attempts) and control (36/48 attempts) hearts; instead, hearts returned after pacing to a sinus rhythm that typically manifested as epicardial breakthroughs in ventricles (Supplementary Fig. 18C, top; Supplementary Movie 11). In the remaining attempts, burst pacing induced unsustained arrhythmias of different durations typically caused by one or two propagating waves (and occasionally a distinct reentrant circuit observed in the field of view, Supplementary Fig. 18C, middle; Supplementary Movie 11) that self-terminated after a short period. Two control (but none of patch-implanted) hearts also exhibited sustained (>1 min) arrhythmias caused by a more complex, multi-wave activity (Supplementary Fig. 18C, bottom; Supplementary Movie 11). Overall, histogram analysis (Supplementary Fig. 18D) showed similar frequencies of arrhythmic events in the two groups, with a slightly higher number of shorter episodes induced in patch-implanted hearts and longer episodes induced in control hearts.

Together, these results indicated that after 3 weeks in vivo, epicardially implanted cardiopatches exhibited robust survival and vascularization, maintained electrical function at pre-implantation levels, lacked electrical integration with the host rat heart, and exerted no adverse effects on host electrical properties or vulnerability to arrhythmias.

## Discussion

Compared to cell injection strategies currently tested in clinics, implantation of in vitro engineered functional cardiac tissues could offer several benefits, such as greatly improved survival, retention, and paracrine action of implanted cells at the infarct site, added structural support, and antiarrhythmic effects from full-length coverage of the scar with electrically conducting tissue. While significant progress has been made in generating miniature heart tissue surrogates for drug testing[7,8,12,20], clinical translation of cardiac tissue engineering has faced several challenges, including: (1) inadequate cardiomyocyte maturation, (2) small tissue surface area, (3) small tissue thickness (no perfusable vasculature), and (4) lack of electromechanical integration between implanted and host tissue. In this study, we sought to address the first two challenges by establishing a platform for engineering highly mature and functional human heart tissues (cardiopatches) with a clinically relevant area ($4 \times 4$ cm). This method is rapid (5 weeks from pluripotent state), reproducible across multiple hPSC lines, and does not require the use of electrical or mechanical stimulation, perfusion bioreactors, or other conditions that would complicate future clinical translation. Rather, we

     

utilized a free-floating dynamic culture to enhance nutrient availability[23], flexible frames to support auxotonic tissue loading, and optimized culture media and cell seeding density to develop a highly efficient in vitro cardiac maturation system.

Human cardiopatches engineered on this platform had functional outputs approaching those of adult human myocardium, including the highest reported patch contractile forces (>5 mN for small and >20 mN for large patches), specific forces (>22 mN/mm$^2$), and CVs (~30 cm/s)[6,12,19,46–50]. Importantly, despite significant scale-up in size, the largest engineered cardiopatches exhibited high CVs and uniform cell density, electrical coupling, and propagation across the entire patch that prevented pacing-induced arrhythmias in vitro, contrasting previous reports in the engineered tissues with slow CVs[51]. As recently shown in non-human primates[22,52], large heterogeneous tissue grafts with relatively immature PSC-CMs greatly increased the risk of cardiac arrhythmias, further signifying the importance of engineering spatially uniform and mature functional properties in large tissue patches. Conceivably, the contractile force and CV of cardiopatches could be further increased by ~1.4-fold if randomly oriented CMs were aligned using more elaborate biofabrication[53,54] or bioreactor[13] approaches.

At the cellular level, the force generating capacity per input cardiomyocyte was 8–1400-fold higher for cardiopatches than other engineered human heart tissues[6,46]. This was consistent with ubiquitously observed regular Z-I-H-I-Z-band patterns across sarcomeres, a finding that contrasted previous studies reporting occurrence of H-zones in fewer than a third of the cells[38] or only with high-frequency electrical stimulation[21], or irregular I-bands and absence of H-zones altogether[9,11,12]. Important factors contributing to advanced CM maturation in cardiopatches were sequential application of serum-free followed by serum-containing media and reduced seeding density, which increased functional gene and protein expression as well as cell size, at least in part by upregulating Akt signaling, a known mediator of physiological hypertrophy[55]. Still, some of the advanced ultrastructural features including M-bands and T-tubules were present in a relatively small subset of CMs, which along with the neonatal Cx43 distribution[31] and force–frequency relationship (FFR)[56] warrant future optimization to achieve a fully mature, adult tissue phenotype. Interestingly, across different hPSC lines and culture protocols, cardiopatches with faster twitch relaxation exhibited more positive FFR, consistent with the need for accelerated sarcoplasmic reticulum uptake of Ca$^{2+}$ in frequency-induced CM inotropy[57]. While high-frequency electrical stimulation[58] might improve FFR in cardiopatches, the 83% force levels remaining at 2 Hz stimulation still significantly surpass other reports in the field.

In this study, the in vivo fate of cardiopatches was investigated using two small animal models. The dorsal window chamber model allowed us to, for the first time, monitor survival, vascularization, and function of implanted cardiac tissues in real time in live mice. Successful vascularization and blood perfusion of cardiopatches within 2 weeks of implantation was likely aided by the continued metabolic demand of spontaneously contracting cardiomyocytes[59] that remained functional throughout the study. Robust engraftment and preserved structure and electrical function of implanted cardiopatches were further confirmed using a more clinically relevant and mechanically realistic environment of rat ventricular epicardium. Here we for the first time employed dual optical mapping to simultaneously, with high spatial and temporal resolution, monitor propagation of electrical signals in cardiopatches and recipient hearts and rigorously assess graft's conduction velocity and electrophysiological effects on host epicardium. Compared to previous methods that utilized extracellular recordings[60], topical application of voltage sensitive dyes[61], or comparison of ECG and gCaMP signals[22,52,62], dual mapping allowed tracking of how grafted CMs are activated relative to host CMs including propagation underneath the graft and at the graft-host boundary. We found that while implanted cardiopatches maintained pre-implantation electrical properties, they failed to functionally couple with the recipient hearts and were separated from the epicardium by a thin non-cardiac layer, as observed in previous studies[62,63].

Importantly, we found no adverse paracrine effects of grafted cells on the electrical properties of underlying epicardium and through aggressive burst pacing protocols demonstrated that cardiopatches did not increase vulnerability to arrhythmias in host ventricles. While applying small cardiopatches to infarcted rodent hearts would likely confirm previously reported paracrine benefits on contractile function[18,62–64], large-animal studies are warranted to further evaluate therapeutic safety and efficacy of large cardiopatches towards potential clinical use. Excitingly, Menasche et al. recently demonstrated improved cardiac function in a 68-year-old patient with advanced heart failure following epicardial implantation of a 20 cm$^2$ fibrin-based tissue patch containing hESC-derived SSEA-1$^+$ progenitors[65], providing a foundation for future use of hPSC-based strategies in human heart repair. Tissue patches made of functional hPSC-CMs might further enhance therapeutic benefits, if engineered to be functionally mature, thick, and able to electromechanically integrate with host myocardium. Regardless of patch size and maturity, the remaining challenges (namely thickness and electrical integration) will need to be addressed for the ultimate success of cardiac tissue engineering therapies in clinics.

In conclusion, we have established a scalable methodology to generate the first highly functional human cardiac tissues with clinically relevant dimensions (4 × 4 cm). Together, the relative simplicity of the approach, rapid structural and functional maturation of tissues in vitro, and robust survival, functionality, and vascularization in multiple small animal models provide grounds for further development of this technology towards novel therapies for ischemic heart disease.

## Methods

**Generation and maintenance of hPSCs.** BJ fibroblasts from a healthy male newborn (ATCC cell line, CRL-2522) were reprogrammed episomally into hiPSCs at the Duke University iPSC Core Facility and named DU11 (Duke University clone #11) following verification of pluripotency. RUES2 and H9 hESCs were obtained from and approved for use by Rockefeller University and WiCell Institute, respectively. Cardiomyocytes differentiated from Hes2 hESCs were obtained from VistaGen Therapeutics. hPSCs were maintained as feeder-free cultures on growth factor-reduced Matrigel (Corning, 80 μg/mL or 8.5–10 μg/cm$^2$ coating) in either TeSR-E8 or mTeSR media (Stem Cell Technologies) and passaged as small (10–20 cells) clusters every 4–5 days using 0.5 mM EDTA (1:10–1:40 split ratios) when cells reached 75–85% confluence. With the exception of Fig. 2f, all experiments were performed using DU11 hiPSCs between passages 18 and 45. RUES2 and H9 hESCs were used between passages 60 and 75. All cell lines were routinely tested for Mycoplasma contamination using commercially available kits (MycoAlert, Lonza).

**Cardiac differentiation of hPSCs.** hPSCs were differentiated into CMs via small-molecule-based modulation of Wnt signaling[1,2]. Briefly, feeder-free cultures of RUES2, H9 hESCs, and DU11 hiPSCs were grown to 75–85% confluence and dissociated into single cells using Accutase (Innovative Cell Technologies). We found that cells responded differently to differentiation depending on their maintenance culture media (E8 or mTeSR), which required adjustment of seeding density and small molecule concentration. As such, cells were plated at either 5 × 10$^4$/cm$^2$ (for E8 protocol) or 2 × 10$^5$/cm$^2$ (for mTeSR protocol) with 5 μM Y27632 (ROCK inhibitor, Tocris) and induced either 3 or 2 days after seeding, respectively. Maintenance media was changed daily prior to differentiation. To induce cardiac differentiation (d0), cells were treated with 8–10 μM (for E8 protocol) or 10–14 μM (for mTeSR protocol) CHIR99021 (SelleckChem) in RPMI-1640 with B27(−) insulin (ThermoFisher Scientific). Exactly 24 h later, CHIR was removed and replaced with basal RPMI/B27(−) medium. On d3, half of the old medium was collected and mixed with fresh RPMI/B27(−) medium containing 5μM (final concentration) IWP-4 (Tocris). Gentle swirling of the plate and aspiration of dead cells and debris prior to addition of complete medium improved cell viability. On

d5, IWP-4 was replaced with basal RPMI/B27(−) medium. From d7 onward, cells were fed with RPMI/B27(+)-insulin every 2–3 days, with spontaneous beating generally starting on d7–d10 of differentiation.

**Metabolic selection of hPSC-CMs.** Differentiating CM cultures were purified via metabolic selection between d10 and d12 based on previously described methods[4]. Briefly, cultures were rinsed with PBS and incubated with "no glucose" medium for 48 h (glucose-free RPMI (ThermoFisher Scientific 11879020) supplemented with 4 mM lactate (Sigma L4263), 0.5 mg/mL recombinant human albumin (Sigma A6612), and 213 µg/mL L-ascorbic acid 2-phosphate (Sigma A8960))[3]. Occasionally, cultures required 1–2 days of additional selection to more efficiently eliminate non-CMs. At the end of the selection period, cultures were dissociated into single cells using 0.25% trypsin/EDTA followed by quenching with stop buffer (DMEM, 20% FBS, 20 µg/mL DNAse I (Millipore 260913)) and replated onto fresh Matrigel-coated dishes to remove dead cells and debris.

**Flow cytometry analysis of CM purity.** Dissociated cardiomyocytes were fixed in 1% paraformaldehyde (Electron Microscopy Sciences, EMS) for 15 min at 22 °C, and permeabilized and blocked in FACS buffer (PBS containing 5% chick serum, 0.1% Triton-X, 0.02% sodium azide) for 1 h. Cells were incubated with rabbit anti-cTnT antibody (Supplementary Table 2) for 1 h at 4 °C, washed three times with FACS buffer, and incubated with anti-rabbit Alexa Fluor® 488 antibody (Invitrogen, 1:1000) for 30 min at 22 °C in the dark. After washing, cells were strained through a 30 µm filter and run on a FACSCalibur cytometer (BD Biosciences). Negative controls consisted of undifferentiated hPSCs stained with anti-cTnT. Live single cells were identified and gated based on their forward and side scatter, and cardiomyocytes were gated based on their cTnT expression. Data were analyzed using Flowing Software.

**Cardiopatch fabrication and culture.** To generate 7 × 7 mm 3D human "cardio-patches", 9 × 9 mm polydimethylsiloxane (PDMS, Dow Corning) square molds were microfabricated as previously described[26]. Molds were gas-sterilized and treated with 0.1% pluronic F-127 (Thermo Fisher Scientific) for >1 h to increase the hydrophilic nature of molds (and minimize cell attachment to PDMS) and rinsed with water just prior to use. Nylon frames (9 × 9 mm) (Cerex Advanced Fabrics) were laser-cut, soaked in 70% EtOH to sterilize, and allowed to dry for >30 min prior to transfer into PDMS molds. Hydrogel solution (24 µL human fibrinogen (10 mg/mL, Sigma F4883), 12 µL Matrigel, 24 µL 2x cardiac media (Supplementary Table 1) was mixed with $1 \times 10^6$ (1 MM) or $0.5 \times 10^6$ (0.5 MM) cells in 58 µL cardiac media (Supplementary Table 1). Following addition of 2.4 µL thrombin (50 U/mL, Sigma T7513), cell/gel solution was added to molds and left at 37 °C for 1 h to polymerize (Supplementary Fig. 1). Cardiopatches were removed from molds and cultured in 12-well plates on a rocking platform (GeneMate Rocker, BioExpress) for 21 days in 1.5–2 mL of cardiac medium, which contained aminocaproic acid to prevent fibrin degradation. To enhance the paracrine action of secreted factors, 2/3 of the culture media was changed every 2 days.

To generate scaled up cardiopatches, 18 × 18 mm Mega and 41 × 41 mm Giga Teflon masters (negative) were first designed in AutoCad (Supplementary Fig. 15A, left column), and then cast with PDMS to make reusable, positive molds (Supplementary Fig. 15A, middle columns). Cerex frames were laser-cut to fit tightly into the molds. Scaled up patches are referred to by their inner frame dimensions (15 × 15 mm for Mega, 36 × 36 mm for Giga; Supplementary Fig. 15A, right column). Hydrogel solution was scaled-up in proportion to the surface area (fold changes from control molds: ~3.5 for Mega, ~17 for Giga), with similar increases in input cell numbers. Giga and Mega master molds were engineered with 12 PDMS posts (Supplementary Fig. 15A, 2nd column) to facilitate eventual removal of the frame for implantation. At a minimum, four small corner posts were required for Giga patches (Supplementary Fig. 15A, 3rd column; Supplementary Fig. 15B) to secure the large Cerex frame in the mold during pipetting and polymerization of the hydrogel mixture. Mega and Giga patches were kept in the molds for 2–3 days to allow for sufficient compaction of the gel across the larger area prior to removal from the molds. While Mega patches could be cultured in six-well plates, Giga patches were cultured in custom-built high-walled PDMS chambers (Supplementary Fig. 15C) to allow for dynamic culture without media spillage or loss of sterility.

**Assessment of electrical propagation.** Optical mapping of transmembrane potentials was performed after 1–3 weeks of culture using our established methods[25,66,67]. Briefly, hPSC-CM patches were incubated with a voltage-sensitive dye, di-4-ANEPPS (15 µM, Life Technologies), in standard Tyrode's solution (135 mM NaCl, 5.4 mM KCl, 1.8 mM CaCl₂, 1 mM MgCl₂, 0.33 mM NaHPO₄, 5 mM HEPES, 5 mM glucose; pH 7.4, 280 mOsm), and a bipolar platinum point-electrode was used to stimulate (8–15 V) the corner of the patch at varying pacing rates (1–4 Hz). Blebbistatin (5 µM, Sigma B0560) was added to inhibit contractions and eliminate motion artifacts during recordings. Two-second episodes of electrical activity induced by point stimulation were recorded from underneath for control and Mega patches using a 504-channel photodiode array (RedShirt Imaging, 1mm effective resolution, acquired at 1.2 kHz) or from above for Giga patches using a fast EMCCD camera (iXonEM+, Andor) equipped with a 50 mm Navitar lens

(512 × 512 pixels, 80 µm resolution, acquired at 125 Hz; Supplementary Fig. 16C). Velocity of action potential propagation (conduction velocity, CV), action potential duration at 80% repolarization (APD), isochrone maps and movies of action potential propagation were derived from acquired signals using our custom MATLAB software[66].

**Assessment of biomechanical properties.** Force generating capacity of cardiac tissue patches was assessed in 1- to 3-week-old patches loaded into a custom-made isometric force measurement setup containing an optical force transducer (µN-sensitivity) and a computer-controlled linear actuator (Thorlabs), as previously described[25,67,68]. To derive force–length relationships, cardiopatch frames were cut on two of four sides and patches were progressively stretched in increments of 2% of culture length (0.14 mm/0.3 mm/0.72 mm for control/Mega/Giga; Supplementary Fig. 16A, B) to a maximum 20% stretch, and at each length, passive tension (non-stimulated) and active (contractile) force responses were recorded during 1 Hz field-electrode stimulation (10 ms duration, 20–30 V) applied by a Grass stimulator (SD9, Grass Technologies). Stiffness was measured as the slope of the passive tension curve at the highest three strain levels (112–120%) divided by the cross-sectional area of the patch (same as the one used for calculation of specific forces). Kinetic properties of contractile forces generation were determined by measuring force rise time (from 10 to 90% activation), decay time (from 10 to 90% relaxation), and total time (10% activation to 90% relaxation) with custom MATLAB algorithms as shown previously[24]. Force–frequency relationship was assessed in normal Tyrode's solution at 112% of culture length via a 20 s recording with step-wise increases in field-shock stimulation at 1, 1.5, and 2 Hz.

**Structural characterization and immunofluorescence in vitro.** Immuno-fluorescent analysis was performed after 1–3 weeks of culture as previously described[24,25]. Briefly, cardiopatches were washed with Ca²⁺-free PBS and fixed with cold 4% paraformaldehyde (EMS) for 15 min on a rocker. Tissues were blocked and permeabilized in 3D block solution (PBS, 0.5% Triton-X100, 5% chicken serum) for 2–3 h at room temperature or overnight at 4 °C, and incubated with 1° antibodies (Supplementary Table 2) overnight at 4 °C. After washing, tissues were incubated with species-appropriate AlexaFluor (Invitrogen) secondary antibodies (1:1000 dilution) overnight at 4 °C. Tissues were mounted on microscope slides in Fluoromount-G (EMS), covered with a coverglass and sealed with nail polish for long-term preservation of fluorescence. Images were taken on a Leica SP5 inverted confocal microscope and post processed with ImageJ.

**Immunofluorescent analysis ex vivo.** Structural analysis of implanted (dorsal window chamber and rat heart) cardiopatches was performed in one of two ways: (1) whole-mount staining, similar to in vitro protocol above, and (2) cryosectioning with subsequent immunostaining. Notably, dorsal window chamber-bearing mice were euthanized via isoflurane inhalation and aortic transection, and full-thickness dorsal skin regions containing cardiopatches were immediately cut out under cold-Tyrode's solution, rinsed with PBS and fixed with 2% PFA for 24–48 h at 4 °C. Skin/patch explants were allowed to equilibrate in 30% sucrose solution for 1–3 days at 4 °C, embedded in OCT, slowly frozen on liquid nitrogen, and cut into 10 µm sections on a cryostat. For immunostaining, frozen sections were rinsed with PBS to remove OCT, blocked with 2D block solution (PBS, 0.2% Triton-X, 5% chick serum) for 2–3 h at room temperature, incubated with 1° antibodies (Supplementary Table 2) diluted in PBS + 5% chick serum overnight at 4 °C, and then 2° antibodies diluted similarly for 1 h. A single drop of Fluoromount-G was added onto each section and then covered with a coverglass, sealed with nail polish and imaged on an Axio Observer fluorescent microscope. For epicardially implanted patches, rat hearts were directly immersed in OCT after ex vivo imaging/mapping, slowly frozen on liquid nitrogen, and cut into 10 µm sections on a cryostat. After rinsing off the OCT, sections were post-fixed with 4% PFA for 15 min at room temperate, and subsequently blocked, immunostained and imaged in the same fashion as sections from dorsal window chamber explants.

**qRT-PCR.** RNA was isolated from 1–3-week-old cardiopatches using a total RNA isolation kit (Bio-Rad). To minimize changes in gene expression during handling of tissues, cardiopatches were flash-frozen immediately in liquid nitrogen after taking out of culture and thawed directly in lysis buffer. For comparison with age-matched 2D monolayers, hiPSC-CMs were plated in parallel onto Matrigel-coated (Corning) 24-well plates at a density of $1 \times 10^5$ cells/cm², cultured in the same medias as cardiopatches (3D RB + d0–7, 5% FBS d7–21), and lysed directly in wells. RNA purity and concentration was measured on a NanoDrop 2000 Spectrophotometer (Thermo Fisher Scientific). RNA was converted into cDNA using the iScript cDNA Synthesis Kit (Bio-Rad). Human gene-specific primers (Supplementary Table 3) were validated using adult left ventricle control samples isolated from healthy males (Duke Human Heart Repository, IRB protocol Pro00005621) and verified to have >85% primer efficiency. qRT-PCR reactions were setup using iTaq SYBR Green Supermix (Bio-Rad) and run on an ABI 7900HT Fast Real-Time PCR system (Applied Biosystems) in triplicates. 5 ng of cDNA was run per well of a 384-well plate using 10 µL reactions. Relative expression of target genes was quantified by the ΔΔCt method using GAPDH as the housekeeping gene.

**Western blotting and protein/DNA ratio**. To isolate protein from tissues, 3-week-old cardiopatches were cut with scissors in 100–120 μL of ice-cold RIPA lysis buffer (Thermo Fisher) containing protease inhibitor (Sigma), vortexed periodically for 2–3 h, and centrifuged at 15,000×g for 30 min to remove the insoluble fraction. When appropriate, phosphatase inhibitor (Cocktail 3, Sigma) was added to lysis buffer to prevent de-phosphorylation. Protein from input cell population (d0 of cardiopatch) was isolated by pelleting dissociated cells, resuspending in lysis solution, and analogous high-speed centrifugation. Protein concentration was measured with a Pierce BCA Protein Assay (Thermo Fisher), with absorbance readings taken at 560 nm. Twenty micrograms of protein was run per lane on a 4–15% Mini-Protean TGX gel (Bio-Rad) and transferred to a 0.2 μm nitrocellulose membrane (Bio-Rad). Membranes were blocked in TBS containing 0.1% Tween-20 and 5% milk, carefully cut along desired molecular weight markers, and incubated with primary antibodies (Supplementary Table 2) overnight at 4 °C in custom-sealed plastic bags to minimize antibody use. HRP-conjugated secondary antibodies were incubated for 1 h at room temperature. Chemiluminescence was performed using SuperSignal West Pico Chemiluminescent Substrate (Thermo Fisher Scientific) and imaged using a ChemiDoc MP system (Bio-Rad). Protein bands were quantified using densitometry on Image Lab software (Bio-Rad). To calculate protein/DNA ratios, absolute protein and DNA concentrations were measured from replicate patches. DNA was isolated using a DNeasy Blood & Tissue Kit (Qiagen) and quantified on a NanoDrop 2000 Spectrophotometer (Thermo Fisher Scientific).

**Transmission electron microscopy**. Three-week-old cardiopatches were fixed in 4% gluteraldehyde (EMS) for 1 h at room temperature and stored in 0.1 M phosphate buffer (PB). Tissues were treated with 2% osmium tetroxide diluted in 0.1M PB for 45 min and subsequently dehydrated in solutions with increasing acetone content (30%, 50% 70%, 95%, 100%). Tissues were equilibrated in a 1:1 mixture of acetone and epoxy (Embed 812 resin kit, EMS) overnight, embedded in resin, cured and cut into 60 nm sections using an UltraCut-E microtome (Leica Reichert Jung) equipped with a diamond blade and a water reservoir. Sections were stained with 0.5% uranyl acetate and imaged on a Phillips CM-12 inverted TEM microscope equipped with an XR-60 camera (Advanced Microscopy Techniques).

**Lentivirus production and transduction**. Lentiviral plasmids were constructed from the pRRL-CMV vector (a gift from Dr Inder Verma, Salk Institute). The CMV promoter was substituted by muscle specific promoter MHCK7 driving expression of GCaMP6[45] (pRRL-MHCK7-GCaMP6; Addgene plasmid #65042)[69]. High-titer lentiviruses were produced using second generation lentiviral packaging system. Briefly, 293FT cells (Life Technologies, R700-07) were co-transfected with lentiviral plasmid, packaging plasmid psPAX2 and envelope plasmid pMD2.G (4:2:1 mass ratios) using Lipofectamine 2000 (Life Technologies). Supernatant containing lentiviral particles was collected 72 h after transfection, centrifuged (500×g, 10 min) and filtered through 0.45 mm cellulose acetate filter (Corning) to remove cell debris before combined with Lenti-X Concentrator (Clontech) at 3:1 volume ratio for overnight incubation at 4 °C. Concentrated lentiviral particles were harvested following 45 min centrifugation (1500×g, 4 °C) and resuspended in 1/10 to 1/100 of the original volume in DMEM medium. Plasmids psPAX2 and pMD2.G were obtained from Didier Trono (Addgene plasmids #12260 and #12259). hiPSC-CMs were transduced with MHCK7-gCaMP6 lentivirus following metabolic purification and replating, generally ~d15 of differentiation. Lentiviral concentration was titrated (1:100 to 1:1000 dilution) to minimize toxicity and achieve transduction efficiencies >80% following overnight transduction.

**Implantation into mouse dorsal skinfold window chambers**. All animal experiments were approved by the Duke University IACUC and followed approved ethical practices. Dorsal skinfold window chamber surgeries were done per previously established methods[43,44]. Nude mice (male, ~10 weeks of age; 22–30 g) were anesthetized by intraperitoneal injection of ketamine (100 mg/kg) and xylazine (10 mg/kg). Using sterile techniques, the dorsal skin was attached to a temporary metal "C-frame" along the midline of the back. The skin was perforated in three locations with a 16G needle to accommodate the screws of the chamber, and a circular region (~12 mm) of the forward-facing skin (including cutis, subcutis, retractor and panniculus carnosis muscles, and associated fascia) was cut away to accommodate the window proper. The front and rear pieces of the titanium dorsal skinfold chamber were assembled together from opposite sides of the skin, and hiPSC-CM cardiopatches were laid perpendicular to the intact panniculus carnosis muscle of the rearward-facing skin. A sterile cover glass was placed over the window and secured by placing a stainless-steel retaining ring into the locking grooves of the chamber. At all times during the procedure, exposed muscle/fascia and engineered tissues were superfused with sterile saline solution to prevent from drying. The chamber was secured to the skin by running a mattress suture along the metal frame, and the "C-frame" was removed. Post-operatively, mice were injected subcutaneously with buprenorphine (1 mg/kg) analgesic and allowed to recover on a heating pad.

**Intravital imaging of vasculature and Ca²⁺ transients**. Degree of cardiopatch vascularization within dorsal window chambers was assessed on days 7 and 14 post

implantation. Mice were anesthetized by nose cone inhalation of isoflurane and placed on a heating pad under a microscope objective. Hyperspectral brightfield image sequences (10 nm increments from 500 to 600 nm) were captured at ×5 magnification using a tunable filter (Cambridge Research & Instrumentation, Inc.) and a DVC camera (ThorLabs), as previously described[43,44]. A custom MATLAB (MathWorks) script was applied to create maps of total hemoglobin concentration, which were further processed using local contrast enhancement in ImageJ (CLAHE plugin, FIJI) and thresholded to binary images to identify vessel area and calculate blood vessel density (BVD, total area of blood vessels per patch area; Supplementary Fig. 17).

Spontaneous Ca²⁺ transients were recorded in real-time immediately after imaging of blood vessels while mice were still under anesthesia. Fluorescent gCaMP6 signals in implanted patches were imaged through a FITC-filter using a fast fluorescent camera (Andor, at 16 μm spatial and 20 ms temporal resolution). Amplitudes of spontaneous Ca²⁺ transients were determined using the Solis software (Andor) by averaging relative fluorescence intensity ($dF/F = [F_{peak} - F_{base}]/F_{base}$) from three ~400 × 400 μm² regions within each patch[44].

**Surgical implantation of cardiopatches onto rat hearts**. All animal experiments were approved by the Duke IACUC and followed approved ethical practices. Cardiopatches were subjected to an established pro-survival protocol[70] consisting of a 1 h heat shock treatment 24 h pre-implantation and a 1hr incubation with a cocktail of pro-survival factors (cyclosporine A, IGF-1, pinacidil, Bcl-xL-BH4, and QVD-OPH) immediately prior to surgery. Athymic (nude) rats (male, 10–12 weeks old) were initially anesthetized via nose-cone isoflurane inhalation (3–4%). Under a dissecting microscope, a tracheostomy was performed on animals placed in supine position, and a 16-G blunt needle was inserted into the trachea and connected to a ventilator. Mechanical ventilation (75 bpm, 2.3–3 L/min at 14.7 PSIA (1.0 bar), isoflurane 1.5–2% with supplemental oxygen) maintained adequate sedation for the duration of the procedure. The left lateral thoracic ventral surface was prepped using aseptic techniques, and left thoracotomy was performed through the 3rd intercostal space. After opening of the pericardial sac, cardiopatches were placed onto the anterior epicardial surface and sutured onto the heart with 6-0 polypropylene sutures in two diagonal corners of the patch frame. Cardiopatches were superfused with sterile saline as necessary to ensure adequate hydration prior to closure of the chest cavity, which was performed in a 2 to 3-layer fashion. Rats were maintained on a heating pad and received pre-emptive analgesia in the form of buprenorphine (0.05 mg/kg).

**Imaging of Ca²⁺ transients in implanted cardiopatches**. Rats containing epicardially-implanted cardiopatches were anti-coagulated using an intraperitoneal injection of heparin (5 mg/kg body weight) and anesthetized by isoflurane inhalation. Following sternotomy, the heart was excised and placed in ice-cold Tyrode's solution, and the aorta was cannulated with an 18-G feeding needle and secured with suture thread. The heart was then maintained on a Langendorff perfusion system with 37 °C oxygenated Tyrode's solution, and flow rate was adjusted to maintain a perfusion pressure of 60–80 mmHg. To assess whether each implanted cardiopatch exhibited functional Ca²⁺ transients, the perfused heart was imaged by an EMCCD camera with a 512 × 512 sensor (Andor iXon Ultra 897) and a 50 mm f0.95 TV lens (Navitar) at a sampling rate of 50 Hz. The epicardial surface was illuminated by a 465–495 nm LED light source (SciMedia LEX2) and viewed through a 510–560 nm emission filter to image the cardiopatch-specific gCaMP6 calcium sensor. If spontaneous gCaMP6 flashing was not observed, then the implantation site was probed with a platinum point electrode applying stimuli at a rate of 1 Hz. Hearts that displayed functional patches (either spontaneous or stimulus-induced gCaMP6 signals) were then used for dual mapping of electrical propagation.

**Dual mapping of implanted cardiopatches and rat hearts**. Hearts were labeled with voltage-sensitive dye by slowly injecting 3 ml of 5 μM di-4-ANEPPS into the perfusion line, and the perfusate was changed to Tyrode's solution with 10 μM blebbistatin to prevent motion artifacts. The hearts were simultaneously imaged by two CMOS cameras (MiCAM Ultima, SciMedia) with 150 μm spatial resolution (100 × 100 recording sites) at a 500 Hz sampling rate. Fluorescent emissions of gCaMP6 and di-4-ANEPPS were separated by a dichroic mirror with a 565 nm cutoff, a 510–560 nm emission filter for the gCaMP6 camera, and a >600 nm longpass emission filter for the di-4-ANEPPS camera. To assess electrical coupling of the patch to the heart, data were acquired during normal sinus rhythm and during epicardial point pacing positioned several mm from the patch location. To measure conduction velocity of cardiopatches, data were acquired while applying point stimulus to the patch periphery. Epicardial CVs and APDs of the rat heart were measured underneath the patch area identified by gCaMP6 activity and surrounding the patch and compared to those of control (non-implanted) hearts. Calculation of conduction velocity and action potential duration was performed using custom MATLAB software.

**Assessment of arrhythmogenesis in rat hearts**. To assess host arrhythmogenicity following cardiopatch implantation, age-matched healthy control (no surgery) and cardiopatch-implanted rat hearts were prepped as above, incubated

with blebbistatin and di-4, and subjected to 1 s of burst pacing (2 Hz, and 6–20 Hz in steps of 2 Hz, total of nine episodes) from the base of the ventricle. Electrical activity was measured with a 1-min ECG recording via two electrodes inside the recording chamber, as well as 5 s of optical mapping acquisition around the time of pacing onset. Arrhythmias were determined based on erratic/rapid ECG activity following discontinuation of burst pacing and were verified with optical mapping. Arrhythmias were characterized by duration (e.g. <5 s on ECG) and qualitatively described through voltage propagation movies obtained during optical mapping. Arrhythmias were classified as unsustained if they self-terminated within 1 min following burst pacing, and those longer than 1 min were classified as sustained.

**Statistical analysis**. Data were expressed as mean ± standard error of the mean (SEM), with select data shown as dot-plots to demonstrate variability. Data were tested for normality using the Shapiro–Wilks test. Unless stated otherwise, experiments involving two groups were analyzed with an unpaired Student's $t$-test after ensuring comparable variance among groups. For experiments involving more than two groups, data were analyzed with a one-way ANOVA followed by a post-hoc Tukey's test. Gene expression analysis was performed on log-transformed data and normalized to adult left ventricle controls for each set. Intravital blood vessel density and gCaMP6 d$F/F$ values were analyzed with a two-way repeated-measures ANOVA (followed by post-hoc Tukey's tests) and a paired $t$-test, respectively. Data were analyzed with JMP Pro 13 with a significance level set to $\alpha = 0.05$. Different levels of significance were reported for individual experiments and noted in the figure legends. Sample sizes for in vitro experiments were determined based on variance of previously reported measurements[23,24]. Sample sizes for animal studies were determined, in part, based on cost and animal availability. During fabrication and electromechanical testing of cardiopatches, tissues from various groups were alternated to reduce confounding variables. No randomization of animal groups was necessary for window chamber experiments. Rat heart optical mapping was alternated between control and cardiopatch-implanted hearts. No blinding of animal experiments was done.

**Data availability**. All data supporting the results of these studies are available within the paper, the associated Supplementary Materials, or from the authors upon reasonable request.

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

## Acknowledgements

We thank N. Medvitz, Y. Gao, A. Ganapathi, M. Dewhirst, G. Palmer, H. Li, and S. Okuwa for technical assistance. This study has been supported by Foundation Leducq, NIH grants R01HL104326, R01HL12652, UG3TR002142, and U01HL134764 to N.B. and 5T32GM007171, F30HL122079 to I.Y.S.

## Author contributions

I.Y.S.: experimental design, collection and assembly of data, data analysis and interpretation, manuscript writing. B.W.A.: experimental design, collection and assembly of data, data analysis and interpretation. Y.Q.: experimental design, collection of data. C.P.J.: collection and assembly of data, data interpretation. A.L.C.: collection and assembly of data, data interpretation. M.E.J.: collection and assembly of data, data analysis and interpretation. N.B.: experimental design, financial support, administrative support, manuscript writing, final approval of manuscript.

## Additional information

**Competing interests:** The authors declare no competing financial interests.

