## [Peer Review File · Nature Communications]

Reviewers' comments:

Reviewer #1 (Remarks to the Author):

This manuscript is well written and demonstrates the authors' thought processes and methods. The in vitro work is extensive and describes significant troubleshooting work to optimize the Cardiopatches. The manuscript would be strengthened with use of a cardiac model in vivo to allow better assessment of how these patches would behave in their intended environment.

Specific comments:

1. The authors report that after 3 weeks, the patches formed from hiPSC-CMs had cardiomyocytes surrounded by fibroblasts and smooth muscle cells – please describe where these cell types come from. Were they present in the cell mixture following CM differentiation prior to patch formation or did they only arise after 3 weeks of free-floating dynamic culture?
2. Please discuss potential mechanisms why the timing of change to 5% FBS-containing media might affect results as described.
3. It is not clear why lowering cell density improved cardiomyocyte functional maturation. Is it that there was more room available to allow for hypertrophy leading to improved functional characteristics? Or perhaps there was more nutrient availability (fewer cells to consume) or less acidic pH (fewer metabolizing cells)? Please clarify the proposed mechanism for these findings. In addition, by only testing two densities, it leaves the reader to wonder if another cell density (either lower than 0.5 MM or in between 0.5 and 1 MM) might further improve electrical and mechanical properties.
4. The figure legends for Figures 5B and 5C are missing.
5. Please describe use of gCaMP lentivirus in the methods section.
6. While the dorsal window chamber assay provides a unique opportunity to view vascularization of the graft tissue, it does not provide a realistic environment for cardiac engraftment. Infarcted or ischemic cardiac tissue likely does not have the same capacity for neovascularization of graft tissue as the subdermal space, and it would be difficult to make reliable comparisons between the two models. Similarly, while it is reassuring that the grafts continued to conduct and contract when implanted into nude mice without evidence of arrhythmias, the site of implantation does not provide much prognostic value as one might expect that arrhythmias might originate at the border of the host myocardium with the graft tissue. Given that the grafts in this manuscript were implanted in non-conducting tissue, it is difficult to predict how these might behave when implanted on the heart.

Reviewer #2 (Remarks to the Author):

Shadrin, et al. demonstrated the maturation of iPS cell-derived cardiomyocytes in bioengineered "cardiopatch". They showed structural, molecular, and functional maturation of their cardiopatch. They also could scale up their cardiopatch to clinically meaningful size. The maturation of cardiomyocytes derived from pluripotent stem cells is one of the major remaining hurdles in this field. Overall, the study is well-designed and the data are convincing. However, there are some limitations as follows.

- 1) As the authors say, the future goal of this study is to provide cardiac tissue to repair the injured heart. However, it is not clear how the cardiopatch is utilized to repair the heart. The authors transplanted their cardiopatches on the dorsal skin. Do the authors plan to transplant their

cardiopatch on the surface of the heart in the clinics? If this is the case, I am very keen to see if grafted cardiomyocytes survive with vascularization and electrically integrate with host cardiomyocytes

2) In Figure 1, the authors showed cardiopatch on a rocker turned matured with time-course. However, the mechanism of maturation is unclear. Is a rocker indispensable for the maturation? I feel that the authors should see the maturation without shaking on a rocker in the same time-course.

3) I was surprised to see that mean CV was 28.5 ± 1.0 cm/s but multiple patches had over 40 cm/s (lines 201-202). How many patches did the authors measure the CV in? Showing dot-plot of CV in each condition would help to better understand the variation between patches.

4) The authors claimed that their cardiopatch is "non-arrhythmogenic", but has yet to transplant it into the heart. As mentioned above, I think transplantation study into the heart is required.

5) In Figure 4, the authors did not really show data involving "hypertrophy".

6) In Figure 6, the origin of CD31+ endothelial cells is obscure. Is the antibody against CD31 specific for mouse?

7) Some figure numbers in the text do not seem to correspond to the actual figures. e.g. lines 229, 262, 266-267. Please correct them

Reviewer #3 (Remarks to the Author):

General comments:

This is an impressive study that reports the development of "Mega" and "Giga" cardiopatches that have advanced maturation characteristics in terms of ultrastructural organization, force production, electrical coupling and propagation, gene expression of maturation markers. The patches are evaluated functionally in vitro, and have the highest reported values for specific force and conduction velocity of engineered cardiac tissue. Remarkably, the functional aspects are retained when the cardiopatches are scaled up to sizes as large as 3.6x3.6 cm. Following implantation in mice, the patches retain functionality and become vascularized.

Specific questions and comments:

Line 155 ("physiological force-length relationships (Fig. 2C)"). Only the active force is reported. However, the passive force (and elasticity of the cardiopatch) is also an important mechanical property. What is the relative magnitude of active force compared with passive force? Somewhat related to this, the authors do not report or comment on the force-frequency relation of their cardiopatches, which many in the field also use as a sign of maturation.

Lines 185-186 ("N-cadherin junctions appear to localize at the cell ends (Fig. 3F, right)."). What about connexin-43, which localizes at the cell ends in normal adult ventricular tissue?

Line 238 ("other human cardiac tissues"). Are the authors referring to both engineered tissues and native tissues?

Line 242-3 ("T-tubule-like structures adjacent to Z-discs (Fig. 4K)"). Do the T-tubules reach the point of being organized? Longitudinal sections of the cell using fluorescent labels, e.g. fluorescent WGA (Crossman et al, PLoS One 2011;6e17901) or voltage-sensitive dye (Louch et al, J Physiol

2006;574:519) could answer this question. Additional information could be obtained by determining whether and where Caveolin-3 (needed for biogenesis of T-tubules) is expressed.

Line 246 ("promoted hiPSC-CM hypertrophy"). What is the evidence of hypertrophy? Data was not provided on cell size or other markers of hypertrophy (e.g., signaling pathways associated with physiological hypertrophy, such as PI3K, AKT, AMPK, mTOR).

Lines 262-268. Fig. 6 should be Fig. 5.

Fig. 1F. Staining for Nkx.2.5 at 3 weeks looks to be just as high at 3 weeks as it was at 1 week. This is unexpected in light of the multiple lines of evidence of cellular maturation. Is this a consistent result? The authors should comment on this finding.

Fig. 2D. Why are the isochrones irregularly shaped? One would expect that for isotropic cardiopatches, they would be circular (like in Fig. 5D for the Mega cardiopatch).

Fig. 5D. What accounts for the irregular isochrones for the Giga cardiopatch? Is this typical? They seem indicative of a patchy substrate with poorly coupled cells, but given the larger mapping area of Giga cardiopatches (and lower spatial resolution), one would expect patchiness to become less evident, not more evident. Furthermore, the isochrones for the Mega cardiopatch are much smoother. It would be worth a comment and brief explanation.

Fig. 5H and 5I. Values here are normalized. It would be helpful to provide the average control values of force (mN) and specific force (mN/mm²) in the figure legend or text.

Reviewer #1: We thank the Reviewer for the very helpful comments underlined below. The answers are in normal font. In addition to textual edits related to new results, minor changes throughout the text have been made to improve readability and satisfy formatting requirements of *Nature Communications*.

Comment 1: What is the origin of the fibroblasts and smooth muscle cells that are present in the cardiopatches after 3 weeks of dynamic culture? Were they present in the cell mixture following CM differentiation prior to patch formation or did they only arise after 3 weeks of free-floating dynamic culture?

Response: As shown now in the new Fig. S2A,B, the fibroblasts and smooth muscle cells make the vast majority of non-myocytes typically present following hPSC differentiation and prior to patch formation. Endothelial cells are very rare in the starting cell population (~0.3%, shown in new Fig. S2C) and do not appear to survive to the end of 3-wk cardiopatch culture (new Fig. S3A, lower panels). In contrast, fibroblasts and smooth muscle cells persist in cardiopatches and are predominantly located at the patch periphery (Fig. 1C, S3A upper panel). As we and others have shown (e.g. Zhang et al., *Biomaterials* 2013; Thavandiran et al., *PNAS* 2015), the non-myocytes play critical roles in the formation of functional heart tissues.

Comment 2: Please discuss potential mechanisms why the timing of change to 5% FBS-containing media might affect results as described.

Response: Cardiopatches cultured in 5%FBS media had better contractile function than those cultured in serum-free 3D RB+ media (Fig. 2G,H), consistent with observed improvements in myofibril morphology and N-cadherin localization to cellular junctions (Fig. S6B), expected to aid in the generation and transmission of active force. These improvements may have come from growth factors present in the serum. However, as FBS is a known mitogen, we now provide data to show a higher abundance of fibroblasts in the serum-containing conditions relative to the 3D RB+ media (new Fig. S6C). As such, we believe that increased proliferation of non-myocytes led to the observed decrease in active force with time of serum exposure, thus counteracting the positive effects on CM maturation. Since non-myocytes reside primarily at the cardiopatch periphery (Fig. 1C, S3A), they likely did not physically interfere with electrical coupling of CMs, which also appeared to increase with serum exposure (Fig. S6A) contributing to higher CVs (Fig. 2I).

Comment 3: It is not clear why lowering cell density improved cardiomyocyte functional maturation. Is it that there was more room available to allow for hypertrophy leading to improved functional characteristics? Or perhaps there was more nutrient availability (fewer cells to

consume) or less acidic pH (fewer metabolizing cells)? Please clarify the proposed mechanism for these findings. In addition, by only testing two densities, it leaves the reader to wonder if another cell density (either lower than 0.5 MM or in between 0.5 and 1 MM) might further improve electrical and mechanical properties.

Response: Our rationale was that lowering initial cell density would both provide more room for cardiomyocyte growth and facilitate mass transfer. If these conditions still allowed the formation of a uniform and dense CM syncytium, cardiopatch function could be improved. After 3 weeks of culture, morphometric analysis of cardiopatches suggested that decreasing the seeding density led to an increase in cell size (54% of cells in 71% of the volume in 0.5MM vs. 1MM patches, or a ~30% larger cell volume). This has been further supported by new results showing higher total protein/DNA ratio (new Fig. 4C) and GAPDH/LamB1 protein ratio (Fig. 4D,E) in 0.5MM vs. 1MM cardiopatches. We also performed Western blot analysis that revealed upregulated Akt signaling in 0.5MM patches (new Fig. S12), suggestive of increased hypertrophic signaling through the PI3K/Akt pathway, possibly due to increased access to growth factors. Finally, expression of functional proteins such as SAA, β MHC, and Cx43 were up to 1.7-fold higher per cell in the 0.5MM condition (Fig. 4D,E), suggestive of cellular maturation and consistent with improved electrical and mechanical function. Furthermore, from our experience, lower seeding densities (e.g. 0.3MM) are not sufficient to support the formation of dense and uniform functional syncytium throughout the patch volume, thus worsening the overall cardiopatch performance.

Comment 4: The figure legends for Figures 5B and 5C are missing.

Response: Thank you for noticing this omission. Appropriate figure legends have been added.

Comment 5: Please describe use of gCaMP lentivirus in the methods section.

Response: The methods section now describes in detail use of gCaMP6 lentivirus.

Comment 6: While the dorsal window chamber assay provides a unique opportunity to view vascularization of the graft tissue, it does not provide a realistic environment for cardiac engraftment. Infarcted or ischemic cardiac tissue likely does not have the same capacity for neovascularization of graft tissue as the subdermal space, and it would be difficult to make reliable comparisons between the two models. Similarly, while it is reassuring that the grafts continued to conduct and contract when implanted into nude mice without evidence of arrhythmias, the site of implantation does not provide much prognostic value as one might expect that arrhythmias might originate at the border of the host myocardium with the graft tissue. Given that the grafts in this manuscript were implanted in non-conducting tissue, it is difficult to predict how these might behave when implanted on the heart.

Response: We thank the reviewer for this important comment. We utilized the window chamber environment for our *in vivo* experiments because it: 1) allows real-time, non-invasive visualization/assessment of implanted patches and 2) resembles epicardial environment with respect to the existence of only one host-implant interface available for vascular ingrowth and diffusion (the other being impermeable glass window). In our previous study (Juhás et al., PNAS 2014), we showed that engineered tissues implanted in window chambers undergo ischemic stress/damage before the host vessel ingrowth and perfusion are established. Owing to their relatively small thickness, initially avascular cardiopatches were still able to robustly survive, vascularize, and continue to function within this environment. Nevertheless, we agree with the reviewer that unlike inside the window chambers, cardiopatches implanted on the epicardium may: 1) be less likely to survive and vascularize due to vigorous motion of the heart surface and

2) induce adverse electrophysiological effects on host cardiomyocytes (directly or via paracrine effects) and increase vulnerability to arrhythmias.

We have thus performed epicardial implantation of 2-wk old cardiopatches in healthy adult nude rats and after 3 weeks *in vivo* investigated their survival, vascularization, and function, as well as their ability to functionally integrate with host myocardium and alter its electrical function and vulnerability to arrhythmias. We performed these studies in healthy hearts because: 1) we feel that the ability of human cardiopatches to improve function of infarcted hearts should be studied in large animals (porcine, non-human primates) rather than rodents that have physiology vastly different from that of humans, which was out of scope of current study; 2) chances for functional electrical coupling to occur between the implanted patch and the epicardium are likely the highest in healthy hearts; 3) arrhythmia vulnerability is low (but not 0) in healthy hearts, thus providing conditions to more readily demonstrate potential arrhythmogenic effects of implanted patches; and 4) animal numbers needed to reach statistical significance are smaller for healthy hearts. Importantly, in these studies we for the first time employed dual CMOS camera optical mapping to simultaneously, with high spatial and temporal resolution, monitor propagation of electrical signals in both cardiopatches (transduced with gCaMP6) and recipient hearts (stained with voltage-sensitive dye Di-4-ANEPPS). This allowed us to rigorously assess graft's electrical properties as well as electrophysiological effects on host epicardium. Compared to previous methods that utilized extracellular recordings (Zimmermann et al., Nature Medicine 2006), topical application of voltage sensitive dyes (Weinberger et al., Sci Transl Med 2016; Furuta et al., Circ Res 2006), or comparison of ECG and gCaMP signals (Chong et al., Nature 2014; Shiba et al., Nature 2016), dual mapping in our studies allowed tracking of how grafted CMs are activated relative to host CMs including propagation underneath the graft and at the graft-host boundary.

Results of these studies are presented in detail in the new Fig. 7, Fig. S18 and Videos S10-11 and are summarized below. Upon heart extraction, 10/11 implanted cardiopatches exhibited gCaMP6-reported Ca^{2+} transients indicating successful engraftment and survival. Dual optical mapping of patch-implanted, Langendorff perfused hearts demonstrated that cardiopatches maintained electrical properties relative to those pre-implantation. However, in no studied hearts we found evidence for functional electrical coupling between the patch and heart since spontaneous or pacing-induced heart activity did not induce patch activation and rare spontaneous beats of the patch did not activate heart, while simultaneous activation of both implanted cardiopatch and underlying epicardium by a stimulus electrode caused the electrical propagation in both patch and heart. From histological analyses, we found that patches maintained dense structure and contained striated and coupled hiPSC-CMs as well as blood vessels originating from the host myocardium. Consistent with lack of graft-host electrical coupling, we observed presence of an insulating non-cardiac layer (200-300 μ m) separating the cardiopatch and epicardium, consistent with previous reports (Gerbin et al., PlosOne 2015; Yildirim et al., Circulation 2007; Weinberger et al., Sci Transl Med 2016).

While these studies ruled out the direct effects of patch on heart via functional electrical coupling, we further assessed electrical properties of epicardium underneath and around the implanted patch as well as in control non-implanted nude rat hearts, and found no evidence for altered host electrical properties (CV, APD) by paracrine effects of the graft. We then systematically assessed in 6 patch-implanted and 6 control Langendorff-perfused hearts the incidence of arrhythmias by applying an aggressive programmed pacing protocol along with ECG recordings and dual optical mapping. We found similar incidence of arrhythmic events in the two groups, with a slightly higher number of shorter arrhythmia episodes induced in patch-implanted hearts and longer episodes induced in control hearts. Taken together, similar as in the window chamber studies, epicardially implanted cardiopatches survived, vascularized, and preserved their pre-implantation electrical function. They neither electrically integrated with host

hearts nor showed adverse paracrine effects on the heart's electrical properties or vulnerability to arrhythmias. These results warrant future studies to address the lack of host-graft functional integration as one of the main challenges for the ultimate success of cardiac tissue engineering therapies in clinics.

Reviewer #2: We thank the Reviewer for the very helpful comments underlined below. The answers are in normal font. When shown, text from the revised manuscript is given in quotes and italics. In addition to textual edits related to new results, minor changes throughout the text have been made to improve readability and satisfy formatting requirements of *Nature Communications*.

Comment 1: As the authors say, the future goal of this study is to provide cardiac tissue to repair the injured heart. However, it is not clear how the cardiopatch is utilized to repair the heart. The authors transplanted their cardiopatches on the dorsal skin. Do the authors plan to transplant their cardiopatch on the surface of the heart in the clinics? If this is the case, I am very keen to see if grafted cardiomyocytes survive with vascularization and electrically integrate with host cardiomyocytes.

Response: We thank the reviewer for this important comment. We have now implanted cardiopatches on the rat epicardium and assessed their survival and vascularization as well as electrical integration and arrhythmogenicity by first-time using dual camera mapping to simultaneously monitor electrical propagation in graft and host cardiomyocytes. The results of these studies are described in the new Figures 7 and S18 and Videos S10-11. Please see the response to the Comment 6 of the reviewer #1 for the specifics of the obtained results.

Comment 2: In Figure 1, the authors showed cardiopatch on a rocker turned matured with time-course. However, the mechanism of maturation is unclear. Is a rocker indispensable for the maturation? I feel that the authors should see the maturation without shaking on a rocker in the same time-course.

Response: We now provide results comparing the structure and function of cardiopatches cultured for 3 weeks in free-floating dynamic conditions (on a rocker) vs. those cultured traditionally in static dishes. Compared to dynamic culture, statically cultured cardiopatches exhibited ~4.9-fold lower forces (1.03 vs. 5.1mN), ~3-fold lower CVs (8.7 vs. 27.2 cm/s), less organized sarcomeric structures, and decreased Cx43 expression (new Fig. S10). Overall, 3-week statically cultured cardiopatches structurally and functionally resembled 1-week dynamically cultured cardiopatches (Figs. 1H, S3B, 2B,E), suggestive of inferior maturation under static conditions. These results agree with our recent study showing that compared to static culture, dynamic culture improves maturation and function of 3D cylindrical microtissues made of neonatal rat or human cardiomyocytes, in part via effects on mTOR signaling (Jackman et al., Biomaterials 2016).

Comment 3: I was surprised to see that mean CV was 28.5 ± 1.0 cm/s but multiple patches had over 40 cm/s (lines 201-202). How many patches did the authors measure the CV in? Showing dot-plot of CV in each condition would help to better understand the variation between patches.

Response: The mean CV results in Fig. 3H are an average of n=38 and 34 patches for 1MM and 0.5MM groups, respectively, which makes the standard errors rather small (~1.0 cm/s). Per reviewer's suggestion, we now show all functional data in revised Figs. 3G,H as dot-plots with sample numbers specified in the figure legend. Within these plots, one can see 3 patches with CVs over 40cm/s in the 0.5MM condition.

Comment 4: The authors claimed that their cardiopatch is "non-arrhythmogenic", but has yet to transplant it into the heart. As mentioned above, I think transplantation study into the heart is required.

Response: We thank the reviewer for this important comment. Please see our response to the Comment 1.

Comment 5: In Figure 4, the authors did not really show data involving “hypertrophy”.

Response: By “hypertrophy” in the title of the Figure 4, we wanted to refer to an increase in cell size. Specifically, after 3 weeks of culture, morphometric analysis of cardiopatches in Fig. 3B-E suggested that decreasing the seeding density led to an increase in cell size (54% of cells in 71% of the volume in 0.5MM vs. 1MM patches, or a ~30% larger cell volume). This has been further supported by new results showing higher total protein/DNA ratio (new Fig. 4C) and GAPDH/Lamb1 protein ratio (Fig. 4D,E) in 0.5MM vs. 1MM cardiopatches. We also performed Western blot analysis that revealed upregulated Akt signaling in 0.5MM patches (new Fig. S12), suggestive of increased hypertrophic signaling through the PI3K/Akt pathway, possibly due to increased access to growth factors. Finally, to remove potential confusion, the Figure 4 legend has been rewritten as: “*Increased cell maturation and size in low-density cardiopatches*”. Please see also the response to the Comment 3 of the reviewer #1.

Comment 6: In Figure 6, the origin of CD31+ endothelial cells is obscure. Is the antibody against CD31 specific for mouse?

Response: While the CD31 antibody is not specific for mouse, we now show in the new Fig. S2C and S3A lower panels, that endothelial cells were very rare in the starting cell population (~0.3%, Fig. S2C) and did not appear to survive 3-wk cardiopatch culture (Fig. S3A). Thus, cardiopatches were avascular at the time of implantation, and the most likely source of the CD31⁺ capillary structures formed *in vivo* were host mouse (window chamber implantation) or rat (epicardial implantation) endothelial cells. This notion is further supported by the presence of human-specific HNA staining only in the patch but not in capillary structures spanning from the host epicardium to cardiopatch shown in the new Fig. 7G. Please see also the response to the Comment 1 of the reviewer #1.

Comment 7: Some figure numbers in the text do not seem to correspond to the actual figures. e.g. lines 229, 262, 266-267. Please correct them.

Response: We apologize for this omission. Figure references have been corrected throughout the text.

Reviewer #3: We thank Reviewer for the very helpful comments underlined below. The answers are in normal font. When shown, text from the revised manuscript is given in quotes and italics. In addition to textual edits related to new results, minor changes throughout the text have been made to improve readability and satisfy formatting requirements of *Nature Communications*.

Comment 1a: Line 155 ("physiological force-length relationships (Fig. 2C)"). Only the active force is reported. However, the passive force (and elasticity of the cardiopatch) is also an important mechanical property. What is the relative magnitude of active force compared with passive force?

Response: We now show (new Fig. 2C) the passive tension-length relationships at 1, 2, and 3 weeks of cardiopatch culture. Passive tension increased with tissue stretch and time of culture with ~2:1-3:1 active:passive force ratios found at the higher stretch levels (Fig. 2C). Given their cross-sectional area, 3-week old cardiopatches exhibited a passive stiffness (average slope of passive stress-strain relationship at 12-20% strain) of ~26kPa, comparable with reports for diastolic stiffness of adult human ventricle (20-50kPa; Neagoe et al., *J Muscle Res & Mot*, 2003; Chen et al., *Mat Sci and Eng*, 2008; Fan et al., *J Biomech Eng*, 2015).

Comment 1b: Somewhat related to this, the authors do not report or comment on the force-frequency relation of their cardiopatches, which many in the field also use as a sign of maturation.

Response: As shown in new Fig. S5, some cardiopatches exhibited slightly positive, some exhibited flat, and most exhibited negative force-frequency relation (FFR). In average, the FFR was slightly negative with $95 \pm 0.8\%$ and $83 \pm 2.0\%$ of the 1Hz force magnitude maintained at 1.5Hz and 2Hz stimulation rates, respectively, (Fig. S5A-B). Interestingly, across different hPSC lines, differentiations, and culture protocols (n=61 patches), the FFR slopes were higher in cardiopatches with shorter 1 Hz twitch duration (Fig. S5C) and, specifically, in those having shorter 1 Hz twitch relaxation (Fig. S5D) but not rise (Fig. S5E) times. This is consistent with the need for accelerated sarcoplasmic reticulum uptake of Ca^{2+} in frequency-induced CM inotropy. While we mentioned negative FFR as one of the neonatal cardiopatch features that require future optimization, the 83% force levels remaining in cardiopatches at 2Hz stimulation rate still significantly surpass functional outputs of any other reported human engineered heart tissues.

Comment 2: Lines 185-186 ("N-cadherin junctions appear to localize at the cell ends (Fig. 3F, right)"). What about connexin-43, which localizes at the cell ends in normal adult ventricular tissues?

Response: Connexin-43 in 3-week old cardiopatches localized uniformly in cell membrane (Fig. 3F) rather than at the cell ends. In normal postnatal development in rodents and humans, polarization of N-cadherin at the cell ends precedes polarization of Cx43 (Vreeker et al., *PloS One*, 2014; Angst et al., *Circ Res*, 1997). A description of the Cx43 localization relative to that of N-cadherin has been now added to the revised Results. Furthermore, in the revised Discussion, uniform rather than polarized Cx43 distribution was further mentioned as one of the neonatal cardiopatch features that will require future optimization towards achieving adult phenotype.

Comment 3: Line 238 ("other human cardiac tissues"). Are the authors referring to both engineered tissues and native tissues?

Response: We apologize for the confusion. The comparison was made with other *engineered* human cardiac tissues. This has been clarified in the text.

Comment 4: Line 242-3 ("T-tubule-like structures adjacent to Z-discs (Fig. 4K)"). Do the T-tubules reach the point of being organized? Longitudinal sections of the cell using fluorescent labels, e.g. fluorescent WGA (Crossman et al, PLoS One 2011;6e17901) or voltage-sensitive dye (Louch et al, J Physiol 2006;574:519) could answer this question. Additional information could be obtained by determining whether and where Caveolin-3 (needed for biogenesis of T-tubules) is expressed.

Response: We thank the reviewer for this comment. We have performed additional analysis of TEM images for the presence of T-tubules in 3wk cardiopatches and found that a minority of cells (<5%) expresses T-tubular structures. In new Fig. S14, we further present Western blot analysis for T-tubule-associated proteins Caveolin-3 and Junctophilin-2 showing a significant upregulation in cardiopatch culture as well as immunostainings showing evidence for a punctate distribution of Cav3 co-localizing with CM cross-striations. Simultaneously, live membrane staining of Di-8-ANEPPS in cardiopatches did not reveal cross-striated pattern of T-tubules characteristic of adult cardiomyocytes. Overall, while we see multiple evidence for T-tubulogenesis after 3 weeks of cardiopatch culture, future optimization will be needed to achieve the mature adult phenotype. Results and Discussion have been revised to include these new studies.

Comment 5: Line 246 ("promoted hiPSC-CM hypertrophy"). What is the evidence of hypertrophy? Data was not provided on cell size or other markers of hypertrophy (e.g., signaling pathways associated with physiological hypertrophy, such as PI3K, AKT, AMPK, mTOR).

Response: We appreciate this comment that complements the comments made by the other two reviewers. We have now addressed this comment in the text and in new Fig. 4C and S12 and revised Fig. 4D-E. Please see our response to the Comment 3 of the reviewer #1 and Comment 5 of the reviewer #2.

Comment 6: Lines 262-268. Fig. 6 should be Fig. 5.

Response: Correct figure references are now provided in the text. We apologize for the omission.

Comment 7: Fig. 1F. Staining for Nkx2.5 at 3 weeks looks to be just as high at 3 weeks as it was at 1 week. This is unexpected in light of the multiple lines of evidence of cellular maturation. Is this a consistent result? The authors should comment on this finding.

Response: Nkx2.5 was used in our study to specifically label hPSC-CM nuclei and quantify expression of proliferation and maturation markers in cardiopatches (Fig. 1F-I). We reassessed our immunostained samples and under the same light intensity and exposure time we can not observe any significant differences in the Nkx2.5 fluorescent intensity among 0,7,14, and 21 days of cardiopatch culture. The antibody for Nkx2.5 is cardiac-specific, as confirmed in the new Fig. S4. Previous work by Lian et al. (PNAS, 2012) utilizing the same base protocol for hPSC-CM differentiation, showed that within first 60 differentiation days, Nkx2.5 gene expression is increased with time of culture (please see their Fig. 3E). Work by others shows that Nkx2.5 is still expressed in the nuclei of adult mammalian myocytes (Kasahara et al., Circ Res 1998) where this transcription factor is believed to play a significant role in maintaining homeostasis and cardioprotection (Akazawa et al., Pharm & Thera 2005). As such, we believe that the expression levels of Nkx2.5 need not necessarily decrease with increased CM maturation, at least over the culture time studied.

Comment 8: Fig. 2D. Why are the isochrones irregularly shaped? One would expect that for isotropic cardiopatches, they would be circular (like in Fig. 5D for the Mega cardiopatch).

Response: We appreciate this concern. As shown in several figures throughout the manuscript and supplement, all 3-wk engineered cardiopatches consist of highly and uniformly packed, randomly oriented, and electromechanically coupled cardiomyocytes. As such, these tissues represent a conducting medium that is functionally isotropic at the macroscopic scale (>1mm), and should ideally exhibit circular action potential spread upon point stimulation. In optical mapping experiments, how regular and circular are the displayed activation isochrones depends on various factors, including the level of noise in recorded signals, spatial and temporal resolution of recordings, the size/geometry of the tissue, proximity of recording sites to stimulus site, stimulus strength, and electrode geometry. We agree with the reviewer that activation isochrones in the 3-wk control (7x7mm) cardiopatches appear less circular (curved) compared to isochrones in Mega (as well as Giga) cardiopatches. There are two likely explanations for this: 1) Bipolar point stimulation will launch action potential that subsequently propagates to develop into a circular wavefront. Compared to small patches, in larger patches, there is more tissue area for the circular waveform to fully develop as well as for modulating effects of electrical stimulation to decay away. 2) The wavefront launched at the corner of the patch will be accelerated along the patch boundaries (that have reduced downstream load), which in turn will decrease the wave curvature in the patch center. Since these effects also decay away from the patch boundary, in the larger cardiopatches the circularity of the wavefront will be less affected by the presence of tissue boundaries than in the small patches.

Furthermore, we still see a slightly higher level of irregularity in isochrones from 1 wk cardiopatches that decreased with time of culture (Fig. 2D). This could be attributed to initial random differences in cell distribution causing areas of slightly slower or faster conduction, but as the cardiomyocytes and intercellular coupling matured with time of culture, the conduction became uniform throughout the entire patch.

Comment 9: Fig. 5D. What accounts for the irregular isochrones for the Giga cardiopatch? Is this typical? They seem indicative of a patchy substrate with poorly coupled cells, but given the larger mapping area of Giga cardiopatches (and lower spatial resolution), one would expect patchiness to become less evident, not more evident. Furthermore, the isochrones for the Mega cardiopatch are much smoother. It would be worth a comment and brief explanation.

Response:

The apparent “roughness” of the displayed activation isochrones primarily depends on the noise level and spatial and temporal resolution of processed signals used to generate isochrones maps. Noise levels across the recorded area are determined by intrinsic noise of the recording device, the type (temporal, spatial, or both) of filtering and degree of filtering, uniformity of voltage sensitive dye staining, and uniformity of illumination pattern. As detailed in the supplementary methods section, action potentials in control (7x7mm) and Mega cardiopatches were recorded using a photodiode array (PDA) connected to a bundle of 504 hexagonally arranged optical fibers (19.5 mm longest inter-vertex distance) at 1X magnification in contact fluorescence imaging mode using trans-illumination (the original method described in Entcheva et al., J. Cardiovasc. Electrophysiology, 2000; Bursac et al., Circulation Research, 2002). Temporal resolution of these recordings was 1.2 KHz and effective spatial resolution was ~1.1 mm. With the intrinsically low noise level, large pixel area, high temporal resolution that provides more sampling points during action potential upstroke to allow more accurate derivation of activation times, and hexagonally arranged recording sites that favor more accurate presentation of circular isochrones, these PDA recordings offer unprecedented quality of signals for derivation of isochrones maps.

To record signals from the entire area of the 40x40mm Giga cardiopatches, we could not use the PDA system (19.5mm field of view), but instead used a fast EMCCD camera (iXonEM+, Andor) with a photographic lens in an epi-illumination mode (Fig. S16C). Compared to the PDA system, the camera system has significantly higher noise level, smaller pixel size (320 μm at used magnification) with a lower temporal resolution used for recordings (125 Hz), and rectangular pixel arrangement, all of which eventually result in a “rougher” appearance of isochrones. Importantly, this does not suggest poor electrical coupling, as the magnitudes of conduction velocities were equal in the 3-wk Giga cardiopatches relative to the 3-wk control and Mega cardiopatches (Fig. 5E). Moreover, similarly irregular isochrones to those shown in Giga cardiopatches were previously presented for healthy hearts in multiple studies (e.g. Tamaddon et al., Circulation Research, 2000; Nygren et al., Am J Physiol Heart Circ Physiol, 2003; Shy et al. Circulation, 2014; Gardner et al., Nature Communications, 2015).

Comment 10: Fig. 5H and 5I. Values here are normalized. It would be helpful to provide the average control values of force (mN) and specific force (mN/mm^2) in the figure legend or text.

Response: We updated these studies to ensure that presenting normalized values for direct comparison among control, Mega, and Giga cardiopatches was done for the same batches of cells. Furthermore, absolute average values for cardiopatches of different sizes are now provided in the revised text.

REVIEWERS' COMMENTS:

Reviewer #1 (Remarks to the Author):

The authors have satisfactorily addressed the points raised by reviewers in the previous round. The addition of an in vivo cardiac model to this manuscript significantly increases the strength of the paper and the relevance to clinical translation. While there are still some limitations, namely lack of electrical integration and use of a non-injury model, the authors are careful not to overstate their findings and implications.

Reviewer #2 (Remarks to the Author):

Shadrin, et. al. worked very hard to answer criticisms raised by the reviewers and I am happy to say that the manuscript has been substantially improved. In particular, epicardial transplantation of cardiopatches and subsequent dual optical mapping study are very impressive. I have no further comments.

Reviewer #3 (Remarks to the Author):

The authors have addressed all of the questions and comments of my original review. I just have a few minor questions/suggestions concerning the new experiments concerning the cardiopatches on rat heart.

a. Lines 370-72 state there is "no evidence of graft-host functional coupling since spontaneous or electrically induced activation in epicardium did not yield Ca^{2+} transients in the patch (Fig. 7D)". While this appears to be the case for spontaneous activity, it seems that electrical pacing resulted in synchronous action potentials and calcium transients (in the signals labeled p in the right panel). Isn't this consistent with (perhaps, intermittent) functional coupling?

b. Line 372. Video S11 should be Video S10.

c. Line 379. For clarity, perhaps change Fig. 7E to Fig. 7E, left.

Responses to remaining reviewers' comments

Reviewer #1:

Comment: The authors have satisfactorily addressed the points raised by reviewers in the previous round. The addition of an in vivo cardiac model to this manuscript significantly increases the strength of the paper and the relevance to clinical translation. While there are still some limitations, namely lack of electrical integration and use of a non-injury model, the authors are careful not to overstate their findings and implications.

Response: We thank the reviewer for the insightful comments. Lack of electrical integration between the implanted patch and recipient heart indeed remains a great challenge in cardiac tissue engineering field. Studies in our group are currently under way to find possible solutions to this problem.

Reviewer #2:

Comment: Shadrin, et. al. worked very hard to answer criticisms raised by the reviewers and I am happy to say that the manuscript has been substantially improved. In particular, epicardial transplantation of cardiopatches and subsequent dual optical mapping study are very impressive. I have no further comments.

Response: We appreciate reviewer's feedback that helped improve the revised manuscript.

Reviewer #3:

Comment 1: The authors have addressed all of the questions and comments of my original review. I just have a few minor questions/suggestions concerning the new experiments concerning the cardiopatches on rat heart.

Response: We sincerely appreciate the detailed questions of this reviewer throughout the revision process. Answers to remaining questions are provided below.

Comment 2: Lines 370-72 state there is "no evidence of graft-host functional coupling since spontaneous or electrically induced activation in epicardium did not yield Ca²⁺ transients in the patch (Fig. 7D)". While this appears to be the case for spontaneous activity, it seems that electrical pacing resulted in synchronous action potentials and calcium transients (in the signals labeled p in the right panel). Isn't this consistent with (perhaps, intermittent) functional coupling?

Response: We thank the reviewer for this question. As already noted in the main text, the lack of calcium transients in the cardiopatch during spontaneous heart activation is, indeed, indicative of absent antegrade coupling (host to donor). Additionally, spontaneous calcium transients from the cardiopatch did not yield action potentials in the heart, suggesting a lack of retrograde coupling (donor to host). As for the case of electrical pacing, the pacing electrode was placed at the edge of the cardiopatch yielding simultaneous activation of both the cardiopatch and the underlying host heart. Thus, even in this case, we could not conclude that the patch and heart were electrically coupled. To address the reviewer's comment, we better clarified in the main text, figure, and movie legend the position of electrode during

pacing, either remote from the patch yielding activation of the heart but not patch, or at the edge of the patch yielding simultaneous activation of both patch and heart.

Comment 3: Line 372. Video S11 should be Video S10.

Response: We thank the reviewer for noticing this omission. The video citation has been corrected.

Comment 4: Line 379. For clarity, perhaps change Fig. 7E to Fig. 7E, left.

Response: We appreciate the suggestion. The figure citation has been adjusted as requested.